# A CD4[+] T cell reference map delineates subtype-specific adaptation during acute and chronic viral infections

**Massimo Andreatta[1,2,3], Ariel Tjitropranoto[4], Zachary Sherman[4], Michael C Kelly[5], Thomas Ciucci[4,6]\*[†], Santiago J Carmona[1,2,3]\*[†]**

[1]Department of Oncology, UNIL CHUV and Ludwig Institute for Cancer Research Lausanne, University of Lausanne, Lausanne, Switzerland; [2]Agora Cancer Research Center, Lausanne, Switzerland; [3]Swiss Institute of Bioinformatics, Lausanne, Switzerland; [4]David H. Smith Center for Vaccine Biology and Immunology, Department of Microbiology and Immunology, University of Rochester, Rochester, United States; [5]Single Cell Analysis Facility, Frederick National Laboratory for Cancer Research, Leidos Biomedical Research Inc, Frederick, United States; [6]Laboratory of Immune Cell Biology, Center for Cancer Research, National Cancer Institute, National Institutes of Health, Bethesda, United States

**\*For correspondence:**
thomasciucci@icloud.com (TC);
Santiago.Carmona@unil.ch (SJC)

[†]These authors contributed equally to this work

**Competing interest:** The authors declare that no competing interests exist.

**Abstract** CD4[+] T cells are critical orchestrators of immune responses against a large variety of pathogens, including viruses. While multiple CD4[+] T cell subtypes and their key transcriptional regulators have been identified, there is a lack of consistent definition for CD4[+] T cell transcriptional states. In addition, the progressive changes affecting CD4[+] T cell subtypes during and after immune responses remain poorly defined. Using single-cell transcriptomics, we characterized the diversity of CD4[+] T cells responding to self-resolving and chronic viral infections in mice. We built a comprehensive map of virus-specific CD4[+] T cells and their evolution over time, and identified six major cell states consistently observed in acute and chronic infections. During the course of acute infections, T cell composition progressively changed from effector to memory states, with subtype-specific gene modules and kinetics. Conversely, in persistent infections T cells acquired distinct, chronicity-associated programs. By single-cell T cell receptor (TCR) analysis, we characterized the clonal structure of virus-specific CD4[+] T cells across individuals. Virus-specific CD4[+] T cell responses were essentially private across individuals and most T cells differentiated into both Tfh and Th1 subtypes irrespective of their TCR. Finally, we showed that our CD4[+] T cell map can be used as a reference to accurately interpret cell states in external single-cell datasets across tissues and disease models. Overall, this study describes a previously unappreciated level of adaptation of the transcriptional states of CD4[+] T cells responding to viruses and provides a new computational resource for CD4[+] T cell analysis.

## Editor's evaluation

This paper uses single-cell genomics to examine the heterogeneity of virus-specific CD4 T cells over time in both acute and chronic viral infection. Further, the authors build a comprehensive atlas of the transcriptional evolution of virus-specific CD4 T cell responses that could be used as a reference tool to interpret other datasets. This work characterizes how the antiviral CD4 T cell transcriptional landscape changes with time and will be of broad interest to those that study acute and chronic CD4 T cell responses.

## Introduction

CD4[+] T cells play a critical role in shaping immune responses against pathogens through the secretion of soluble mediators and direct cell interactions with other immune cell populations. The multifaceted ability of CD4[+] T cells to orchestrate multiple layers of protection relies on their unique capacity to adopt diverse functional fates upon antigen encounters (*Nguyen et al., 2019*; *Swain et al., 2012*). Following viral infections, naive CD4[+] T cells clonally expand and differentiate into effector populations supporting both cellular and humoral responses. This functional diversification of CD4[+] T cell populations is under tight transcriptional control, ensuring the appropriate positioning and deployment of effector functions (*Zhu et al., 2010*). While Th1 cells, supported by the transcription factors Blimp1 and T-bet, regulate cellular responses in helping CD8[+] T cells and innate populations through the secretion of INF-γ, follicular-helper CD4[+] T cells (Tfh), which depend on Bcl6, promote antibody responses via direct cell contact with B cells and the production of cytokines like IL-21 (*Crotty, 2011*; *Laidlaw et al., 2016*; *Sheikh and Groom, 2021*).

During the contraction phase following its initial amplification, the pool of virus-specific CD4[+] T cells declines in both self-resolving and persistent infections. However, the nature of the infection greatly impacts the evolution of CD4[+] T cell functions (*Brooks et al., 2005*; *Crawford et al., 2014*; *Fahey et al., 2011*). After acute viral infections, pathogen clearance is followed by the persistence of memory CD4[+] T cell populations that acquire distinct phenotypes, gene expression and functional properties (*Crawford et al., 2014*; *Hale et al., 2013*; *Marshall et al., 2011*). Because memory populations are heterogeneous and include many subsets, including Th1- and Tfh-like subsets, measuring transcriptional changes occurring between the effector to memory populations remains challenging. In fact, in addition of Th1 and Tfh cells, memory populations are comprised of a less differentiated subset of Central Memory (Tcm) cells that contribute to long-term protective functions of CD4[+] T cells (*Pepper and Jenkins, 2011*). Tcm cells phenotypically resemble cells present during the early anti-viral response and referred to as Central Memory precursors (Tcmp), raising the possibility of an early transcriptional imprinting that favors the emergence of long-lived memory CD4[+] T cells (*Ciucci et al., 2019*; *Marshall et al., 2011*; *Pepper et al., 2011*). Yet, the nature of such program, as well as its overlap with that involved in Th1 and Tfh subsets, has not been elucidated. More broadly, it remains unclear whether a shared transcriptional module regulates memory differentiation, or whether diverse gene programs allowing long-term maintenance are imprinted in a subset-specific manner. In sharp contrast to acute settings, chronic infections do not result in such phenotypic memory transition of persisting cells (*Brooks et al., 2005*; *Fahey et al., 2011*). Instead, in response to sustained antigenic stimulation, CD4[+] T cells acquire dysfunctional features, including the expression of inhibitory receptors and reduced cytokine production (*Brooks et al., 2005*; *Crawford et al., 2014*). In this context, questions remain about how persistent infections alter the functional and transcriptional landscape of CD4[+] T cell populations. However, because of the lack of consistent definition of virus-specific CD4[+] T cell states across conditions and over time, the subtype-specific adaptations during infections are currently poorly characterized.

Although there is evidence that cell fate decisions are stochastically imprinted on T cells (*Buchholz et al., 2016*; *Buchholz et al., 2013*; *Soon et al., 2020*), other studies have shown that cell-intrinsic factors as well as environmental cues affect the differentiation of single naïve T cells (*Cho et al., 2017*; *Tubo et al., 2016*; *Tubo et al., 2013*). Among these factors, the interaction between the T cell receptor (TCR) and their cognate antigen has been shown to influence the diversification and maintenance of CD4[+] T cells (*Cho et al., 2017*; *Snook et al., 2018*; *Tubo et al., 2013*). For instance, recent studies showed that Th1 and Tfh differentiation are influenced by TCR usage, affinity to cognate peptides and the type of infection (*Khatun et al., 2021*; *Künzli et al., 2021*; *Snook et al., 2018*). Yet, we do not fully understand the extent by which the TCR repertoire impacts the early fate decision of CD4[+] T cells responding to acute and chronic viral infections. In particular, it remains to be addressed whether naïve T cells with particular TCR chains are preferentially recruited during the effector phase and adopt specific transcriptional profiles that could skew the overall immune response.

Here, we employed single-cell RNA sequencing (scRNA-seq) coupled with single-cell TCR sequencing to explore the landscape of virus-specific CD4[+] T cell states at different timepoints during acute and chronic infections. We provide evidence of both shared and subtype-specific transcriptional changes occurring dynamically in both types of infection. Analysis of paired scRNA-seq and scTCR-seq data of antigen-specific polyclonal T cells revealed that, although a fraction of the clonotypes in both

acute and chronic settings were significantly biased towards specific subtypes, T cell functional diversification appears to be mostly independent of the expression of particular TCR chain pairs. Based on these results, we make available a new reference map describing virus-specific CD4[+] T cell states – including Th1, Tfh, and Tcmp/Tcm – and their dynamic evolution over time during acute and chronic infection. By combining this map with a reference-projection algorithm, we provide a new computational framework that enables automated and accurate interpretation of CD4[+] T cell states across models, conditions, and experiments.

## Results

### Differential phenotypic adaptation of CD4[+] T cells in acute and chronic viral infection

To characterize the diversification of T cell populations during acute and chronic infections, we used two variants of the lymphocytic choriomeningitis virus (LCMV): the Armstrong and Clone 13 strains. Both viruses induce a strong T cell amplification early in the response, followed by the persistence of a small pool of virus-specific T cells at later timepoints. While the Armstrong strain results in an acute infection cleared within 6–8 days post infection (dpi), Clone 13 persists, leading to chronic infection (*Ahmed et al., 1984*; *Crawford et al., 2014*). Using these models, we sought to measure the phenotypic changes of virus-specific T cells following infection, both at early and late timepoints. Virus-specific CD4[+] and CD8[+] T cells were identified using MHC tetramers loaded with the LCMV-derived GP66 and GP33 peptides, respectively. Cells were analyzed using spectral flow cytometry with a panel of 21 parameters allowing high-dimensional analyses based on 14 surface markers expressed by virus-specific T cells (*Figure 1A*, *Figure 1—figure supplement 1A-C*).

Cytometric readouts showed that splenic virus-specific CD4[+] T cells are highly heterogeneous and largely differ phenotypically from CD8[+] T cells, both at early and late timepoints after acute and chronic infections (*Figure 1B*, *Figure 1—figure supplement 1D*). Interestingly, while CD4[+] T cells responding to acute and chronic infections at early timepoints showed partial similarities, minimal overlap was observed at late timepoints. Additionally, CD4[+] T cells differentiating in chronic settings appear to change less drastically over time compared to the sharp transition occurring between early and late timepoints after acute infection (*Figure 1C*). Overall, these analyses revealed a profound and fast-adapting phenotypic heterogeneity of CD4[+] T cell populations in response to different infection settings.

### Defining the landscape of CD4[+] T cell states in acute and chronic viral infection

To characterize the transcriptional landscape of virus-specific CD4[+] T cells and gain further insight into their heterogeneity and transcriptional adaptation, we conducted single-cell RNA sequencing (scRNA-seq) of virus-specific CD4[+] T cells isolated from LCMV-infected animals at different timepoints. Virus-specific GP66:I-A[b+] were purified either 7 or 21 days after Clone 13 infection – conditions referred to as Early and Late Chronic. In addition, similar populations were isolated 7, 21 and >60 days after LCMV Armstrong infection – conditions referred to as Acute, Early and Late Memory, respectively. A total of 11 samples, including two or three biological replicates per condition, were processed with droplet-based scRNA-seq, resulting in over 35,000 high-quality virus-specific CD4[+] T cell transcriptomes (*Figure 2A* and *Supplementary file 1A-B*). In addition, selected samples were used to measure simultaneously TCR usage and transcriptome at single-cell resolution. To generate a unified map of virus-specific CD4[+] T cell states in acute and chronic infections, all datasets were integrated with STACAS, a computational tool allowing correction of batch effects while preserving relevant biological variability across datasets (*Andreatta and Carmona, 2021a*) (see Materials and methods) (*Figure 2B*). While different timepoints and types of infection (i.e. acute vs chronic) occupied different areas of the integrated space, biological replicates were largely covering overlapping areas of the map (*Figure 2—figure supplement 1A*), suggesting a successful data integration. By clustering the high-dimensional space of the integrated map, we defined six major and three minor CD4[+] T cell clusters that were annotated based on the expression of canonical markers previously described in this model. Among the six major clusters, representing >93% of the cells in the map, we identified: *(i)* Th1 effector cells, expressing the highest levels of *Cxcr6* and *Ly6c2* (encoding Ly6C); *(ii)* Tfh effector cells,

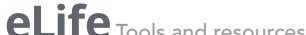

**Figure 1.** Phenotypic characterization of virus-specific CD4+ T cells by spectral flow cytometry. Spleen GP66:I-A^b+ CD4+ T cells were analyzed 7 and 21 days after infection with LCMV Armstrong and Clone 13. (**A**) Schematic of experimental procedures. Uniform Manifold Approximation and Projection (UMAP) visualization was calculated based on the expression of 14 markers on virus-specific CD4+ T cells pooled from 5 animals. (**B**) Expression of selected markers shown on the UMAP as in (A). (**C**) CD4+ T cells from each condition were highlighted as contour lines on the UMAP. Experiment with 4–5 mice per group, representative of two independent experiments. See also *Figure 1—figure supplement 1* for spectral cytometry analysis including GP33:H2D^b+ CD8+ T cells, and for surface marker panels used to characterize T cell populations.

The online version of this article includes the following figure supplement(s) for figure 1:

**Figure supplement 1.** Phenotypic characterization of virus-specific T cells.

preferentially expressing *Cxcr5* and *Izumo1r* (encoding FR4); (*iii*) Central memory precursors (Tcmp) expressing the highest levels of *Ccr7* (*Figure 2C*). The remaining three major clusters were identified as putative memory populations based on their higher expression of memory-associated genes *Tcf7* (encoding TCF1) and *Il7r*, corresponding to: (*iv*) Th1 memory (co-expressing *Tcf7*, *Il7r*, *Cxcr6* and *Ly6c2*), (*v*) Tfh memory (co-expressing *Tcf7*, *Il7r* and *Izumo1r*) and (*vi*) Central Memory cells (Tcm), with the highest levels of *Tcf7* and *Il7r* but limited expression of Th1/Tfh marker genes (*Figure 2C*). Consistent with these annotations, Th1 memory, Tfh memory and Central Memory (Tcm) populations were predominantly derived from virus-specific CD4+ T cells isolated at late timepoints after acute infection (*Figure 2—figure supplement 1A*, see next section). In addition, these clusters of virus-specific CD4+

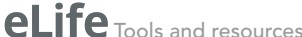

**Figure 2.** Transcriptional landscape of CD4+ T cell states during infections. (**A**) Schematic experimental design to assess virus-specific T cell transcriptomes at different timepoints in acute (Armstrong) and chronic (Clone 13) LCMV infections (additional information in *Supplementary file 1*). (**B**) UMAP visualization of single-cell data before and after dataset integration, highlighting the samples and batches on the left and the 9 CD4+ T cell subtypes of the reference map on the right. (**C**) Expression levels of key marker genes in the 9 subtypes of the reference map. (**D**) Single-cell expression visualized in the UMAP space for key marker genes of Th1, Tfh and Tcmp/Tcm subtypes. (**E**) Average expression of differentially expressed genes in the six major subtypes of the reference map; selected genes are highlighted.

The online version of this article includes the following figure supplement(s) for figure 2:

**Figure supplement 1.** Distribution and consistency of CD4+ T cell states during infections.

T cells and their annotations were independently validated using gene signatures that were previously identified on CD4+ T cells after acute viral infection (*Ciucci et al., 2019*; *Figure 2—figure supplement 1B*). Finally, three minor clusters corresponded to (*i*) *Foxp3*-expressing regulatory T cells (Treg), (*ii*) a Tfh-like state expressing high levels of type 1 interferons-stimulated genes (INFI-stimulated) and (*iii*) a population characterized by high levels of *Eomes* (Eomes-HI) (*Figure 2C*). These minor states

were largely associated to chronic infection and will be described in a later section. All subtypes were present in similar proportions across biological replicates, and samples clustered by condition rather than by batch, further confirming a successful data integration (*Figure 2—figure supplement 1D-E*). Similar subtype proportions were confirmed by spectral cytometry (*Figure 1—figure supplement 1E*).

We next explored the expression of genes involved in the function of CD4+ T cell subsets across major clusters. As expected, Th1 cells, both effector and memory, expressed the highest amount of the Th1-defining transcription factor *Prdm1* (encoding Blimp1) together with *Gzmb* – encoding the cytotoxic effector molecule Granzyme B. Similarly, the Tfh-specific transcription factor *Bcl6* and effector molecule *Il21* were almost exclusively expressed in Tfh cells (*Figure 2D*). Interestingly, Tcmp and Tcm populations, which expressed the highest levels of *Ccr7*, *Il7r* and *S1pr1* were characterized by the promiscuous expression of both Th1-associated genes such as *Nkg7*, *Ifngr1*, or *Runx3*, and Tfh-specific markers like *Slamf6*, *Tox*, or *Bcl2* (*Figure 2C–D*, *Figure 2—figure supplement 1C*).

Differential gene expression analysis between the major CD4+ T cell states revealed additional subtype-specific genes, and showed that Th1 states (both Effector and Memory subsets) share a common gene module (e.g. *Id2*, *Runx3*, *Gzmb*), distinct to that of Tfh cells, characterized by the expression of *Tox*, *Maf*, and *Izumo1r* (*Figure 2E*). Although Tcmp and Tcm states are largely defined by their lack of Th1 and Tfh gene signatures, consistent with a more quiescent, undifferentiated state, they are characterized by the shared expression of genes such as *Ccr7*, *Klf2*, and *S1pr1*. The full list of CD4+ T cell subtype-specific signatures is available in *Supplementary file 2*, and gene expression in this dataset can be explored online at https://spica.unil.ch/refs/viral-CD4-T.

## Subtype-specific evolution of CD4+ T cell states during acute infection

To further describe CD4+ T cell states as they adapt during the course of an acute, self-resolving infection that generates protective memory T cells, we investigated subtype composition and subtype-specific transcriptional changes over time. First, from the effector phase (7 dpi) to the early memory phase (21 dpi), we observed a dramatic shift from Th1 effector, Tcmp and Tfh effector to Th1 memory, Tcm and Tfh memory (*Figure 3A*), concomitantly associated with a reduction in the absolute number of T cells (*Figure 2—figure supplement 1F-G*). This was consistent with the fact that major functional and phenotypic changes occurs during and after the contraction phase 12–20 days following infection (*Marshall et al., 2011*; *Pepper and Jenkins, 2011*).

To further investigate the transcriptional changes underlying this transition, we sought to identify gene expression differences among matching cell states during the acute response and early memory phase. We also interrogated potential changes between the early and late memory time-points resulting in the identification of 196 genes differentially expressed in a time- and state-specific manner in response to acute infection (*Figure 3—figure supplement 1A*, *Supplementary file 2*). Our analyses revealed that the transition of Th1 and Tfh subtypes from effector phase to memory phase were accompanied by the dampening of effector function molecules such as *Gzmb* (Th1) and *Il21* (Tfh), and by the acquisition of *Il7r* expression at the memory phase (*Figure 3B*). However, the down-regulation of effector programs, especially for the Tfh state, was more pronounced in late memory phase compared to early memory, suggesting that memory CD4+ T cells undergo continued transcriptional remodeling after the contraction phase. In contrast, the central memory-type cells [Tcmp and Tcm clusters, referred to as Tcm(p)] readily downregulated most effector-associated genes at the early memory phase. Similarly, the Tcm(p) state more quickly upregulated genes associated with the function, survival or trafficking of memory cells like *Ccr7*, *Il7r*, *Bcl2*, or *Klf6* compared to Th1 and Tfh states. This early divergence and stable expression of memory genes is compatible with the possibility that Tcmp represent a pool of circulatory cells with an increased fitness to develop into long-lived memory cells.

These analyses highlight subtype-specific transcriptional changes from effector to memory states. In particular, they suggest that, while Tcmp are poised to transition to memory, Th1 and Tfh states do so with different kinetics and using divergent transcriptional modules.

## Subtype-specific adaptation of CD4+ T cell states to chronic infection

We next investigated the adaptation of CD4+ T cell transcriptional states to chronic infection. Compared to the acute setting, we did not observe a sharp transition to memory states at day 21, and

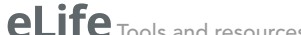

**Figure 3.** Subset-specific adaptation of CD4⁺ T cell states in chronic and acute infections. (**A,C**) Distribution of T cell states at different timepoints after acute and chronic infections. In the UMAP plots, contour lines indicate the density of T cells for each type of infection and timepoint; the barplots in the bottom row indicate the percentage of cells in each subtype in the indicated condition. (**B,D**) Normalized average expression during acute and chronic infections among Th1, Tfh and Tcm(p) subtypes. Selected genes from differentially expressed genes shown in *Figure 3—figure supplement 1A,C*.

*Figure 3 continued on next page*

*Figure 3 continued*

(**E,F**) Animal were infected with LCMV Armstrong (Acute) or Clone 13 (Chronic) and analyzed at the indicated timepoints (Early: 7dpi; Late: 21dpi). (**E**) Graph shows the percentage of Eomes⁺ cells among spleen GP66:I-A^b+T cells. (**F**) Plot (left) shows the intracellular expression of Eomes and Thpok on spleen GP66:I-A^b+T cells analyzed 21 dpi after LCMV Clone 13 infection. Graph (right) shows Thpok mean fluorescent intensity (gMFI) in the indicated population. (**E,F**) are from one experiment with >5 mice per group, representative of 2 independent experiments.

The online version of this article includes the following figure supplement(s) for figure 3:

**Figure supplement 1.** Adaptation of CD4⁺ T cell states in chronic and acute infections.

most virus-specific CD4⁺ T cells matched effector subtypes (***Figure 3C***). Although the proportion of the Th1 subtype remained similar at early timepoints between acute and chronic condition, there was a reduction in the Th1 effector subsets at late chronic stages. In addition, we observed a larger pool of Tfh cells both at early and late chronic timepoints compared to acute settings (***Figure 3C***)**,** consistent with previous studies highlighting a Tfh bias during Clone 13 infection (***Brooks et al., 2005***; ***Fahey et al., 2011***). We also noted that the fraction of Tcmp cells was lower in chronic infection, both at early and late stages, compared to the CD4⁺ T cells in acute infection. Indeed, the proportion of Tcmp and Tcm cells, which can be identified by flow cytometry based on CCR7 expression, was greatly reduced among virus-specific CD4⁺ T cells in chronic settings compared to acute infection (***Figure 3—figure supplement 1B***).

Next, we aimed to assess the transcriptional changes affecting the differentiation and persistence of each subtype in responses to chronicity. To this end, we measured the differences across subtypes at an early (7 dpi) and late phase (21 dpi) of the chronic response. We identified 214 genes differentially expressed between Acute vs. Early Chronic, Early vs. Late Chronic and Early memory vs. Late Chronic timepoints (***Figure 3—figure supplement 1C***, ***Supplementary file 2***). Importantly, most changes observed at the late chronic phase were not present at the early chronic stage, suggesting that they are not merely attributed to changes in viral replication or host responses to viral variants. We observed that late chronicity was associated with the upregulation of a shared gene module, including Nr4a family members (*Nr4a1, Nr4a2, Nr4a3*) and *Tox* in all subtypes (***Figure 3D***, ***Figure 3—figure supplement 1C***)**,** indicative of the strong TCR engagement in response to persistent antigen (***Seo et al., 2019***). Similarly, the expression of inhibitory receptors such as *Pdcd1* (encoding PD-1) and *Lag3* was also detected across states in late chronic samples. Late timepoints were also characterized by the downregulation of effector modules, including gene associated with cytotoxic function, such as *Gzmb* and *Ctsw* (encoding *Cathepsin W*) in Th1 clusters. In contrast, cytokine *Il21* as well as transcription factor *Maf* remained highly expressed in Tfh clusters, suggesting that, unlike in Th1, effector functions in Tfh cells are not dampened at late chronic phase. In fact, *Il21* expression appears to increase at late timepoints in both Tfh and Th1 subtypes (***Figure 3D***). In contrast to what was observed in response to acute infection, Tcm(p) minimally diverged from other states, as the expression of inhibitory receptors and transcription factors associated with T cell dysfunction such as *Ikzf2* (encoding Helios), *Bhlhe40*, or *Gata3* (***Crawford et al., 2014***; ***Doering et al., 2012***; ***Singer et al., 2016***) were equally upregulated in all states in chronic settings. *However*, unlike other subtypes, Tcm(p) maintained expression of *Il7r*, *Ccr7*, and *S1rp1* (***Figure 3D***, ***Figure 3—figure supplement 1C***).

In addition to the six main CD4⁺ T cell states, we detected two distinct states that were almost exclusively present in response to chronic infection: the IFNI-stimulated state and the *Eomes*-HI state (***Figure 3AC***). The *Eomes*-HI state was specifically observed at late stages of chronic infection, consistent with previous studies (***Crawford et al., 2014***; ***Lewis et al., 2016***) and flow cytometry analyses (***Figure 3E*** **and** ***Figure 3—figure supplement 1D***). Because this cluster was characterized by the co-expression of *Eomes, Lag3* and *Xcl1* (***Figure 3—figure supplement 1E***), three functional targets repressed by the CD4⁺ T cell-defining transcription factor ThPOK (***Ciucci et al., 2019***; ***Taniuchi, 2018***), we sought to determine whether its expression was altered in this subset. Indeed, we found that ThPOK protein expression was reduced specifically in EOMES⁺ virus-specific CD4⁺ T cells late during chronic infection (***Figure 3F***). In addition, this subset displays specific expression of *Crtam* and *Gzmk* (***Figure 3—figure supplement 1E***) and is compatible with the CD4⁺ T cell subtype with high-cytotoxic potential (***Cenerenti et al., 2022***).

In summary, these analyses showed that persistent antigen exposure during chronic infections deeply alters CD4⁺ T cell differentiation by imprinting both common and subtype-specific transcriptional changes associated with chronicity.

## Clonotype-fate relationships of virus-specific CD4+ T cells in acute and chronic infections

Because cell-intrinsic factors, notably the expression of TCR, have been shown to impact the differentiation of virus-specific T cells (*Khatun et al., 2021*; *Künzli et al., 2021*; *Snook et al., 2018*), we sought to determine whether CD4+ T cell states are influenced by the expression of particular sets of TCR chains. To describe the clonal relationship between CD4+ T cell states, we analyzed the TCR usage of all T cells for which a productive pair of TCRα/TCRβ sequences was detected (65% of all single cells). To limit potential confounding factors related to the survival or expansion of clonotypes over time, we restricted our analyses to early timepoints (i.e. 7 dpi) after acute or chronic infections. We observed a consistent pattern of clonal expansion across different animals, with large clonotypes (13–31 clones with >20 cells per animal) occupying roughly half of the clonal space in all samples, both for acute and early chronic settings (*Figure 4A*). Next, we interrogated potential repertoire overlaps between animals considering the pair of nucleotide and amino-acid sequences of the CDR3 regions. Strikingly, this analysis showed that less than 3% of the clonotypes (7 out of 795 CDR3 nucleotide pairs; 20 out of 779 CDR3 protein pairs) were observed in two or more animals, when considering clones with 3 cells or more (*Figure 4B*). Similar results were obtained when all clones, including singletons, were included (*Figure 4—figure supplement 1A*). This suggests that the TCR repertoire of virus-specific CD4+ T cells is largely private, that is subject-specific, even when considering a single peptide specificity and animals with the same genetic background.

Next, we explored the potential relationship between TCR usage and the emergence of specific T cell states at early infection timepoints. To this end, we defined a 'clonotype bias' metric to quantify how individual clones are skewed toward a specific subtype (see Materials and methods). At its extreme values, a clonotype bias of 1 indicates that a clonotype is composed uniquely of cells from the same subtype, and a clonotype bias of zero corresponds to a clonotype that matches exactly the background subtype distribution of the whole sample. Because small clonotypes are statistically more likely to show high clonotype bias compared to large clonotypes, we only considered expanded clonotypes with 10 or more cells, and corrected for clonotype size by generating expected background distributions by random permutation (*Figure 4—figure supplement 1B*, see Materials and methods). This analysis revealed that most clonotypes were largely unbiased, both in acute and chronic infections, giving rise to multiple CD4+ T cell states (*Figure 4C–D*), consistent with previous studies in acute infection (*Khatun et al., 2021*). However, we observed that a small number of clonotypes exhibited significant functional bias (Z-score >5), that is they were preferentially enriched in one specific subtype (*Figure 4C*). In the case of acute infection, 9% of expanded clonotypes (15 out of 165) exhibited a functional bias toward one subtype. Similarly, 13% of expanded clonotypes in chronic infection (11 out of 85) showed a significant clonotype bias (*Figure 4C*). Interestingly, in acute infection 14 out of 15 biased clonotypes were skewed toward either the Tcmp or Th1 states (8 and 6 respectively), while in chronic infection all 11 biased clonotypes showed functional bias toward either Th1 or Tfh states (8 and 3, respectively) (*Figure 4C–E*). We did not observe any robust CDR3 motif associated with biased clonotypes, and previously reported fate-biased CDR3 motifs (*Khatun et al., 2021*) were not predictive of clonotype lineage on our data (*Figure 4—figure supplement 1C-D* and Materials and methods).

These combined analyses of the TCR repertoire and transcriptional landscape reveal that the vast majority of the clonotypes can differentiate into multiple states with minimal functional skewing. However, a minority of these clonotypes are significantly biased toward a particular functional state, and this bias appears to be influenced by the type of infection.

## Reference map projection to dissect the effect of genetic and therapeutic perturbations

A cell atlas is particularly useful when it serves as a 'reference' to compare and interpret new data. We have recently proposed a computational method, ProjecTILs, that allows analyzing single-cell datasets by projection into a reference atlas (*Andreatta et al., 2021c*). Using this approach, we sought to apply our CD4+ T cell reference map for the interpretation of CD4+ T cell states in external datasets (*Figure 5A*).

To verify the accuracy of data projection into our reference map, we analyzed an independent scRNA-seq dataset of LCMV-specific CD4+ T cells isolated at 7 and 30 days post-infection with LCMV

**Figure 4.** Clonal structure of virus-specific CD4+ T cells and clonotype-fate relationship. (**A**) Fraction of clonal space occupied by clonotypes with different levels of expansion. For each sample (three replicates for Acute, two replicates for Early Chronic), the plot indicates the number of single cells that belong to clonotypes in one of the five classes of expansion. (**B**) Number of clones with identical CDR3 nucleotides or amino acid sequence pairs between individual samples among clonotypes with ≥3 cells. (**C**) Clonotype bias analysis for acute and chronic infection samples. Plots show clonotype bias vs. clonal size for all clones with >10 cells. Clonotypes are colored by predominant T cell subtype (left) or by Z-score of the clonotype functional bias (right). Red line highlights the null distribution (background distribution) for each condition. (**D, E**) Distribution of cells over the reference map for the (**D**) four most expanded and (**E**) four most biased clonotypes in acute and chronic infections. The CDR3 alpha and beta sequences and the clonotype size are indicated for each clonotype.

The online version of this article includes the following figure supplement(s) for figure 4:

**Figure supplement 1.** Repertoire overlap of virus-specific CD4+ T cells and clonotype bias.

Armstrong (*Ciucci et al., 2019*). Validating both the accuracy of the map and of the projection algorithm, cells from day 7 were projected into the Tcmp and effector states, while cells from day 30 were largely projected into the memory states (*Figure 5B*). Importantly, the expression profile of key marker genes for the projected samples matched closely to the expression profile of the reference map in all major cell subtypes (*Figure 5C*).

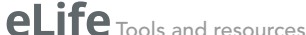

**Figure 5.** Interpretation of CD4+ T cell states in external datasets by projection into the reference map. (**A**) Independent scRNA-seq datasets can be projected into the CD4+ T cell reference map using the ProjecTILs algorithm and interpreted in the context of the space and cell states of the reference. (**B**) UMAP embeddings for the external data by *Ciucci et al., 2019* at 7 and 30 days after acute LCMV infection projected into the reference map. Contour lines indicate the density of projected cells. (**C**) Radar plots showing average expression profiles of a panel of CD4+ T cell marker genes, for

*Figure 5 continued on next page*

*Figure 5 continued*

the projected cells from ***Ciucci et al., 2019*** (blue and red) compared to the reference map profiles (black) grouped by predicted subtype. (**D**) UMAP embeddings for projected scRNA-seq data of Bcl6- and Prdm1-deficient virus-specific CD4$^+$ T cells isolated 7 days after acute infection with LCMV Armstrong (data from ***Ciucci et al., 2022***). (**E**) UMAP embeddings for projected scRNA-seq data of virus-specific CD4$^+$ T cells isolated 33 days after chronic infection with LCMV Clone13, in animals treated with anti-PDL1 or isotype control (data from ***Snell et al., 2021***). Barplots in the bottom row indicate the percentage of cells projected in each reference subtype.

The online version of this article includes the following figure supplement(s) for figure 5:

**Figure supplement 1.** Reference-based classification of independent datasets.

**Figure supplement 2.** Effect of sequencing depth on reference-based annotation.

**Figure supplement 3.** Reference-based analysis of CD4$^+$ T cells in late chronic infection.

For additional validation, we re-analyzed LCMV-specific CD4$^+$ T cells isolated 10 days post-infection with LCMV Armstrong (***Khatun et al., 2021***). Dataset projection into our CD4$^+$ T cell reference map revealed that the majority of these virus-specific T cells were found in the Th1 Effector, Tfh Effector, or Tcmp states (***Figure 5—figure supplement 1A***), similarly to subtype compositions we observed at day 7 in acute infection (***Figure 3A***). This is consistent with the notion that transition to memory phenotypes occurs later, at day 12–20 post-infection (***Marshall et al., 2011***). Importantly, very similar subtype distributions were observed across different mice, highlighting the robustness of the projection algorithm across multiple biological replicates (***Figure 5—figure supplement 1B***). Based on the same dataset, we verified that, while cycling cells tend to cluster together in unsupervised analyses irrespective of their subtype, they are correctly classified by reference projection (***Figure 5—figure supplement 1C-G*** and methods). We also confirmed that data projection is robust to sequencing depth, with consistent subtype annotation with as low as one third of typical sequencing depths (***Figure 5—figure supplement 2*** and methods).

We also tested the robustness of our projection approach in detecting subtle variations across biologically similar samples. To this end, we took advantage of transcriptomic datasets of LCMV-specific CD4$^+$ T cells isolated at memory timepoints (day 35) after in vivo administration of an inhibitor blocking NAD-induced cell death (NICD) (***Künzli et al., 2020***). As expected for a late timepoint, in both the control and treated conditions most cells were projected into the memory clusters (Tcm, Th1 memory, and Tfh memory) (***Figure 5—figure supplement 1H***). However, the NICD-protector-treated sample showed a ~twofold increase of Tfh effector and Tfh memory cells compared to control (***Figure 5—figure supplement 1I***). This is consistent with Tfh cells being more susceptible to NICD than other CD4$^+$ T cell subtypes, and that in vivo NICD-blockade can enhance the persistence of Tfh populations after infection (***Künzli et al., 2020***).

Having validated the accuracy and robustness of reference map projections, we sought to apply the reference map to interpret the effect of genetic and pharmacological perturbations. Transcription factors Bcl6 and Blimp1 (encoded by the *Prdm1* gene) have antagonistic roles in driving CD4$^+$ T cell differentiation into Tfh or Th1 lineages, respectively (***Ciucci et al., 2022***). Projection of scRNA-seq data from genetically altered virus-specific CD4$^+$ T cells isolated 7 days after LCMV acute infection (***Ciucci et al., 2022***) showed that *Bcl6*-deficient cells almost exclusively acquired a Th1 state (Th1 Effector or Th1 Memory), while *Blimp1*-deficient cells were dominated by Tfh and Tcmp states (***Figure 5D***). These results are in line with the known role of Bcl6 in driving Tfh and memory differentiation, with the role of Blimp1 in promoting Th1 functions (***Crotty, 2011***), and highlight the utility of our tool to describe the effect of genetic perturbations.

Next, we aimed at using our reference map to interpret the effect of immunotherapies. To this end, we projected scRNA-seq data of virus-specific CD4$^+$ T cells isolated from mice chronically infected with LMCV, after treatment with an anti-PD-L1 antibody (***Snell et al., 2021***). While control samples showed a similar subtype distribution to our late chronic samples (***Figure 5E, Figure 3C***), anti-PD-L1 treatment increased the relative proportion of Th1 effectors (***Figure 5E***). Expression profiles for all major subtypes in this dataset largely matched those of our reference (***Figure 5—figure supplement 3B***), including the expression of exhaustion markers, which was similar to chronic infection samples of the reference map (***Figure 5—figure supplement 3C***). Notably, Th1 effector cells after anti-PD-L1 treatment upregulated a Th1-associated gene module that includes *Klrd1*, *Plac8*, *Ctla2a*, and *Ly6c2* (***Figure 5—figure supplement 3D***), confirming the findings by ***Snell et al., 2021***.

Overall, these analyses demonstrate that our map and projection method can successfully describe biologically relevant alterations in the subtype distribution and transcriptional programs following diverse perturbations.

## Diversity of virus-specific CD4⁺ T cells across tissues

To investigate whether our reference map can be useful to describe the subtype composition and transcriptional landscape of CD4⁺ T cells from other biological models and tissues, we projected influenza-specific CD4⁺ T cells isolated from lungs and draining lymph nodes (LN) at different time-points after infection (*Swarnalekha et al., 2021*). Data projection revealed that, while lymphoid tissue was largely dominated by Tfh subtypes, lung samples were enriched in Th1 cells, with a progressive accumulation of Tfh cells at later timepoints (*Figure 6A–C*). These results recapitulated recent observations into the delayed development of tissue-resident CD4⁺ T cells with Tfh characteristics in the lungs after influenza infection (*Son et al., 2021*; *Swarnalekha et al., 2021*). In addition to changes in cell subtype composition, we evaluated transcriptional differences between tissues for each subtype. Consistently with the findings by Swarnalekha et al., we observed that, compared to LNs, all the most abundant T cell subtypes in the lung display a tissue-specific gene module, which includes *Crem, Tnfrsf4, H3f3b, Fth1, Ifngr1, Vps37b, Sub1, and Arpc3* (*Figure 6D*).

We next projected LCMV-specific CD4⁺ T cells isolated from liver and spleen at day 37 after acute infection from the study by *Künzli et al., 2020*. As previously reported, liver-resident CD4⁺ T cells were strongly enriched in Th1 subtypes compared to spleen (*Künzli et al., 2020*; *Figure 6E*). Interestingly, the tissue-specific gene module derived from lung in the context of influenza infection was also significantly upregulated by all subtypes of LCMV-specific CD4⁺ T cells in the liver, compared to those from spleen (*Figure 6F*). Altogether, these results suggest that subtype-defining transcriptional programs are preserved across tissues, and that these can be exploited for classification by reference map projection. Moreover, they indicate that different subtypes can use the same gene programs to adapt to different tissues, for instance to acquire residency capacity in non-lymphoid tissue.

## Reference map projection to explore CD4⁺ T cell diversity beyond viral infections

Lastly, we investigated how a transcriptional map of CD4⁺ T cells that developed in response to viral infections could help interpreting the heterogeneity of CD4⁺ T cells differentiating in a non-infectious context. To this end, we isolated and performed scRNA-seq of tumor-specific CD4⁺ T cells from tumor and draining lymph nodes (dLN) of animals inoculated with a colon carcinoma expressing the LCMV-derived GP protein (*Magen et al., 2019*; *Supplementary file 1*). Projection of these data into the reference map showed that most tumor-specific CD4⁺ T cells in the dLN corresponded to Tfh states (*Figure 7A–B*), similarly to virus-specific cells in the LN of infected animals (*Figure 6A–B*). In contrast to viral infection, in tumor-draining LN we observed a sizable fraction (~10%) of Treg among antigen-specific CD4⁺ T cells (*Figure 7A–B*). Moreover, tumor-infiltrating lymphocytes (TILs) were largely projected into the Th1 effector (40–50%) and Treg (~30%) reference map states (*Figure 7A–B*).

Next, we examined each subtype separately. While the gene profile of tumor-specific Tfh were consistent with those of the viral reference, tumor-specific T cells projected into the Th1 effector state seemed to diverge (*Figure 7C*). Indeed, re-clustering and re-calculation of UMAP embeddings (now including both the virus-specific and the tumor-specific T cells) revealed that the T helper cells from the tumor formed a separate cluster (*Figure 7D*). Compared to virus-specific Th1 cells, this tumor-specific CD4⁺ T cell state differentially expressed genes associated with Th2 cells, including *Igfbp7, Ccl1, Ccr8*, and *Il13*, and downregulated Th1-associated genes, including *Ccl5* and *Ly6c2* (*Figure 7E–F*). This suggests that CD4⁺ T cells acquire distinct effector programs in cancer and infection.

While reference maps aim at being as comprehensive as possible, it is possible that new datasets contain novel states that are not represented in the reference, especially when used in different diseases models. In these cases, the user is encouraged to make use of all the analytic tools we provide with the ProjecTILs package (see Materials and methods) to evaluate the degree of correspondence between reference and query, as we illustrated in the case of tumor-specific T cells (*Figure 7C–E*). These analyses demonstrate the feasibility of using a reference map to describe cell diversity beyond the states already present in the map, and as a strategy to expand references to incorporate novel, unrepresented cell states.

**Figure 6.** Diversity of virus-specific CD4+ T cells across tissues. (**A**) Reference projection of influenza-specific CD4+ T cells isolated from draining lymph nodes and lungs at different timepoints after infection (*Swarnalekha et al., 2021*). Black contour lines represent density of cells over the reference UMAP embeddings. (**B**) Summary of subtype composition (percentage of total cells) for each of the six samples in this study. (**C**) Fold-change of cell subtype proportions between lung and lymph node at day 30 p.i. (**D**) Differentially expressed genes for select subtypes between lung and lymph node at day 30 p.i. The genes consistently found in all three comparisons (hereby 'lung residency signature') are: Crem, Tnfrsf4, H3f3b, Fth1, Ifngr1, Vps37b, Sub1, and Arpc3 (p-values from Wilcoxon test). (**E**) Reference projection of LCMV-specific CD4+ T cells from liver and spleen at day 37 p.i. (*Künzli et al., 2020*). (**F**) UCell scores on spleen and liver samples for the "lung residency signature" learned from CD4+ T cells in influenza infection (p-values from Wilcoxon test).

**Figure 7.** Projection of tumor-specific CD4[+] T cells into the reference viral map. (**A**) Reference projection of tumor-specific (GP66:I-A[b+]) CD4[+] T cells isolated from the tumors (TIL) or draining lymph nodes (LN) of animals inoculated with MC38-GP tumor. Black points and contour lines represent projected cells and their density over the reference UMAP embeddings. (**B**) Subtype composition as percentage of total cells for each sample. (**C**) Radar plots showing average expression profiles of a panel of CD4[+] T cell marker genes, for the projected tumor-specific CD4[+] T cells from the indicated organs compared to the reference map profiles (black). (**D**) Re-calculated UMAP plot generated after merging virus-specific T cell data (reference map) with tumor-specific T cell data (projected data). Non-Treg, non-Tfh tumor-infiltrating CD4[+] lymphocytes (TILs) emerge as a distinct cluster ('Tumoral_Th'). (**E**) Differentially expressed genes between virus-specific T helpers (Th1 Effector) and tumor-infiltrating T helpers ('Tumoral_Th'); p-values from Wilcoxon test. (**F**) Expression of genes associated with Th1 or Th2 functions in the indicated cell subtypes (log-normalized UMI counts).

## Discussion

CD4[+] T cells orchestrate immune responses to pathogens and critically support protection conferred by vaccination. However, the phenotypic and functional plasticity of CD4[+] T cells has hindered a robust, unbiased delineation of pathogen-specific T cell subtypes. Although the precise characterization of T cell transcriptional states is fundamental toward understanding the dynamics of immune responses, the subtype-specific changes occurring over time in response to acute and chronic infections remain

poorly understood. In recent years, scRNA-seq has enabled unbiased interrogation of T cell transcriptional diversity at the single-cell level. However, definition of T cell subtypes and states remains subjective and inconsistent across studies. We have previously demonstrated that reference projection is a scalable computational approach for robust and consistent single-cell data analysis, provided a reference cell map (*Andreatta et al., 2021c*).

In this work, we aimed at providing a reference map of the transcriptional and clonal landscape of virus-specific CD4⁺ T cells in acute and chronic infections. To this end, we generated a scRNA-seq dataset of >35,000 high-quality virus-specific polyclonal CD4⁺ T cells from infected mice in multiple timepoints throughout LCMV chronic or acute infections. One key advantage of this model is that it allowed us to explore T cell transcriptional diversity and clonality in both acute and chronic settings using a single antigen-specificity (*GP66:I-A^b+*) and to include biological replicates for every condition and timepoint. Combined with the projection algorithm ProjecTILs, our new reference map enabled robust and consistent interpretation of external CD4⁺ T cell single-cell transcriptomics data from multiple tissues, conditions, and biological models, providing a new powerful resource for the community.

We also presented new insights into the transcriptional adaptation of virus-specific CD4⁺ T cell populations over time and across conditions. Our analyses during acute infection highlighted that, although the major gene expression changes occur at the end of the initial proliferative burst, early memory CD4⁺ T cells that survive the contraction phase undergo continued transcriptional remodeling at later timepoints, similarly to late changes at play in CD8⁺ and NK T cells memory development (*Chang et al., 2014*; *Lau et al., 2018*; *Milner et al., 2020*). However, CD4⁺ T cell subsets undergo memory transition in a divergent manner, where each subtype acquires memory features with different kinetics and using non-overlapping transcriptional modules. Similar to the acquisition of the memory program in CD8⁺ T cells, Th1 memory differentiation is characterized by a dampening of effector functions accompanied by the upregulation of molecules associated with their long-term survival. In contrast, the transition of effector Tfh cells to memory states appears to be delayed, as molecules associated with Tfh function such as IL-21 or ICOS remain expressed in early memory Tfh cells, and their expression only decreases at later timepoints. This delayed transition into resting memory could be associated with differential and prolonged antigen exposure of Tfh within the germinal centers compared to Th1 cells (*Künzli et al., 2020*). Interestingly, the transition to the Tcm state diverges from both Th1 and Tfh. In fact, most memory-associated features like the expression of *Ccr7*, *Il7r*, or *Bcl2* appear quickly at the early memory phase or are already present in Tcmp cells at the acute phase of the response. This observation is in line with the concept that Tcmp cells already express a large fraction of memory-associated genes allowing for their survival and homing to lymphoid organs. Thus, it is possible that most Tcm derived mainly from Tcmp cells through the acquisition of a transcriptional state poised for memory differentiation, similar to memory-precursors in CD8⁺ T cell populations (*Joshi et al., 2007*; *Kaech et al., 2003*).

During chronic infection, similar state-specific changes occur in CD4⁺ T cell subtypes. In addition to a shared transcriptional module upregulated in all subsets, Th1 and Tfh states differ in their adaptation to chronic antigen stimulation. Imprinting of persistent antigen exposure on Th1 cells results in a reduction of effector function characterized by the repression of effector molecules like granzymes, reminiscent of CD8⁺ T cell function dampening in chronic infection and cancer (*Crawford et al., 2014*; *Singer et al., 2016*). In contrast, Tfh effector functions remain unaffected at the chronic phase of the response. While this could be related to the strong Tfh bias observed in chronic infections (*Fahey et al., 2011*), we also observed in Th1 subsets features typically associated with Tfh cells, including the expression of IL-21. Because IL-21 is critical to limit T cell dysfunction during chronic infections (*Elsaesser et al., 2009*; *Fröhlich et al., 2009*; *Yi et al., 2009*), it is possible that compensatory mechanisms enforce its expression in non-Tfh subsets. Alternatively, the 'boundaries' between CD4⁺ T cells states may be less easily delineated during chronic infection. In line with this idea, our analyses reveal that during chronic infection, the accumulation of Eomes⁺ virus-specific CD4⁺ T cells is accompanied by the downregulation of the CD4⁺ T cell-defining factor Thpok. Similar to 'redirected' CD4⁺ T cells in both human and mouse (*Mucida et al., 2013*; *Serroukh et al., 2018*), this CD4⁺ T cell subset is characterized by the upregulation of targets actively repressed by Thpok, including the transcription factor Eomes and genes associated with cytotoxic activity (*Ciucci et al., 2019*; *Vacchio et al., 2019*). While the downregulation of Thpok in CD4⁺ T cells can be influenced by the cytokine milieu

(*Cervantes-Barragan et al., 2017*; *Reis et al., 2014*), it is important to note that the Eomes⁺ Thpok^low CD4⁺ T cell population appears to be limited to chronic settings, including in responses to the gut microbiota (*Cervantes-Barragan et al., 2017*; *Mucida et al., 2013*).

Although we have focused on LCMV viral infection, this study represents the first step into building a more comprehensive reference map of the transcriptional landscape supporting the functional heterogeneity of CD4⁺ T cells across tissues and beyond viral infections. In fact, we have shown that unrepresented transcriptional states (e.g. tumoral Th cells) and programs (e.g. adaptation to non-lymphoid tissue) can be interpreted in the context of the reference map, suggesting a strategy where reference maps can evolve to incorporate novel cell states. We provide user-friendly computational resources for investigators to explore the new CD4⁺ T cell map and to analyze external datasets in the context of this reference. The virus-specific CD4⁺ T cell reference map developed in this study can be explored within the SPICA portal at https://spica.unil.ch/refs/viral-CD4-T, where users can compare the expression of genes of interest in individual cell subtypes and across the reference space. SPICA also hosts interactive analyses for the datasets described above and for several others (https://spica.unil.ch/projects). Finally, researchers can project their own scRNA-seq data through the SPICA web interface, or by using our R package ProjecTILs available at https://github.com/carmonalab/ProjecTILs (*Carmona and Andreatta, 2022*).

## Materials and methods
### Mice, virus, and infections
C57BL/6Ncr were infected by intra-peritoneal injection of $2 \times 10^5$ pfu of LCMV Armstrong or intra-venously with $2 \times 10^6$ pfu of LCMV Clone 13. Viral stocks were prepared and titrated as previously described (*Ciucci et al., 2019*; *Dangi et al., 2020*).

### Antibodies
Antibodies for the following specificities were purchased either from Becton-Dickinson Pharmingen, BioLegend or ThermoFisher: CD4 (GK1.5), CD8α (53-6-7), CD5 (53–7.3), B220 (RA3-6B2), CD44 (IM7), IL-7Ra (A7R34), CCR7 (4B12), CXCR5 (SPRCL5), CXCR6 (SA051D1), PSGL1 (2PH1), Ly6C (HK1.4), CD27 (LG/3A10), FR4 (12A5), Thpok (T43-94), Eomes (Dan11mag), CD69 (H1.2F3), LAG3 (C9B7W), Tim3 (RMT3-23), KLRG1 (2F1), PD1 (29 F.1A12), CX3CR1 (SA011F11). MHC tetramers loaded with the LCMV GP66 or GP33 peptides were obtained from the NIH Tetramer Core Facility.

### Spleen cell preparation and staining
Spleen cells were prepared and stained as previously described (*Ciucci et al., 2019*). Surface staining with GP66:I-Ab tetramer and for CCR7 or CXCR5 was performed at 37 °C for 1 hr prior to staining with antibodies for other cell surface markers. Intracellular stainings were performed as previously described using the Transcription Factor Staining Buffer (ThermoFisher) (*Chopp et al., 2020*). Data was acquired on Aurora spectral flow cytometer (Cytek) and analyzed with FlowJo V10.8 software (TreeStar). Dead cells and doublets were excluded by DAPI or LiveDead staining (Invitrogen) and forward scatter height by width gating. Purification of lymphocytes by cell sorting was performed on a FACS Fusion and FACS Aria (BD Biosciences).

### Tumor model and cell preparation
MC38 colon cancer carcinoma cell line expressing the LCMV-derived GP protein was kindly provided by R. Bosselut at the National Cancer Institute, NIH, and cultured and inoculated as previously described (*Magen et al., 2019*). The cell line was tested negative for *Mycoplasma* by PCR. The cell line is not part of the commonly misidentified cell lines of the International Cell Line Authentication Committee. 2 weeks following subcutaneously injection, tumor and draining lymph nodes were harvested and processed as previously described (*Magen et al., 2019*).

### Single-cell RNA sequencing
GP66:I-A^b+ T cells were sorted from LCMV infected or tumor-bearing animals, loaded onto the Chromium platform (10 X Genomics) to generate cDNAs carrying cell- and transcript-specific barcodes that were used to construct sequencing libraries using the Chromium Single Cell 5' or 3' Library & Gel

Bead Kit according to the manufacturer instructions. For pooled captures, two-cell populations were sorted and barcoded separately with TotalSeq antibodies (BioLegend) before mixing and cell captures (*Supplementary file 1*). Libraries were sequenced on multiple runs of Illumina NextSeq or Novaseq using paired-end reads to reach a sequencing saturation of 60–90%, resulting in at least $2–9 \times 10^4$ reads/cell. Single-cell sequencing files were processed, and count matrixes extracted using the Cell Ranger Single Cell Software Suite (10 X Genomics).

## scRNA-seq and scTCR-seq data processing and quality control

Single-cell transcriptomes and single-cell TCR sequences were mapped and combined using the *combineTCR* function from scRepertoire (*Borcherding et al., 2020*). We performed quality control on the single-cell data using the following criteria: number of detected genes >700; number of UMIs >1500 and<15,000; percentage of ribosomal genes <50 and percentage of mitochondrial genes <10. For all these parameters, we additionally removed all extreme outlier cells outside the 1st and 99th percentile in each sample. In order to filter out potential contaminants and experimental artifacts, we applied the UCell package (*Andreatta and Carmona, 2021b*) to evaluate a panel of signatures for several common immune and non-immune cell types. This resulted in high-quality transcriptomes for 35,488 single cells from 11 samples, covering acute and chronic infections at three different timepoints.

### scRNA-seq data integration

For the construction of the CD4+ T cell reference map, datasets were downsampled to balance the contribution from different types of infection and timepoint. To this end, a maximum of 5000 cells were randomly selected for each of the 5 subsets: acute day 7, acute day 21, acute day 60, chronic day 7 and chronic day 21. To mitigate batch effects between samples, we integrated the 11 samples using STACAS (*Andreatta and Carmona, 2021a*) with the following parameters: *number of variable genes* = 800, *dist.thr*=0.6, dims = 20. For the selection of variable genes for data integration, we did not consider mitochondrial genes, ribosomal genes, heatshock proteins, interferon-stimulated genes (ISGs), cell cycle genes or T cell receptor genes (gene sets available in *Supplementary file 3*). On the integrated data, unsupervised clusters were calculated using the *FindNeighbors* and *FindClusters* functions from Seurat (*Hao et al., 2021*) with parameters: *k.parameter*=10, resolution = 0.4. Finally, unsupervised clusters were manually annotated guided by differential expression analysis between clusters, merging clusters where appropriate, to obtain nine 'functional clusters' that summarize the diversity of CD4+ T cells in acute and chronic infections.

### Dataset projection and reference-based analysis of scRNA-seq data

In order to avoid large imbalances between subtypes, and to limit its disk size, the CD4+ T cells reference map is a downsampled version of all available data. In order to obtain low-dimensional embeddings and subtype annotations for all cells generated in this study, we projected them onto the map using ProjecTILs with default parameters (*Andreatta et al., 2021c*). The same method was applied to project and interpret scRNA-seq data from several additional studies not included in the reference map (*Ciucci et al., 2022*, *Ciucci et al., 2019*; *Khatun et al., 2021*; *Künzli et al., 2020*; *Snell et al., 2021*; *Swarnalekha et al., 2021*). For all re-analyses of public data, we retrieved the gene expression matrices from Gene Expression Omnibus (GEO) under the following accession numbers: LCMV-specific CD4+ T cells at 7 and 30 days p.i. (GSE121002); LCMV-specific CD4+ T cells at day 10 p.i. (GSE158896); LCMV-specific CD4+ T cells at day 35 p.i. (GSE139198); CD4 +T cells from lung and lymph nodes of influenza-infected mice (multiple timepoints), and liver cells at day 37 post-LCMV infection (GSE146626); *Blc6*-deficient, *Blimp-1* deficient and WT CD4 +T cells 7 days post-acute LCMV infection (GSE149912); Anti-PD-L1-treated and isotype-treated virus-specific CD4 +T cells at day 33 p.i. (GSE163345). Prior to reference projection, CD4 +T cells were purified from each dataset using scGate (*Andreatta et al., 2022*) and default T cell and CD4 +T cell gating models.

### Detection of novel/unrepresented states

Several utilities are available in the ProjecTILs package to evaluate query projection accuracy and to detect the presence of novel cell states. First, on a qualitative level, one can apply the *plot.states. radar*() function to compare the expression of panels of key genes between query and reference. Second, the user can recalculate the UMAP embeddings of the combined reference and query space

(function *recalculate.embeddings*()) to assess whether part of the projected data form a novel, separate cluster (e.g. *Figure 7D*). Third, per-subtype differential expression analysis (function *find.discriminant.genes*()) can reveal which and how many genes are differentially expressed between reference and query in a given subtype (e.g. *Figure 7E*). Fourth, the *compute_silhouette*() function calculates an average silhouette coefficient per subtype, which aims at measuring the average distance of query cells from their own assigned cluster compared to all other clusters of the reference. A case study highlighting the application of these metrics can be found at: https://carmonalab.github.io/ProjecTILs_CaseStudies/novelstate.html.

## Gene signatures of adaptation to acute and chronic infections

CD4[+] T cell state signatures were calculated by comparing each of the 3 main effector and 3 main memory clusters to the two other clusters in the same state (i.e. effector or memory) using the *Find-Markers* function implemented in Seurat. To summarize subtype-specific transcriptional changes at different stages of infection, we first identified differentially expressed genes between timepoints and subtypes. For this analysis, Th1 effector and Th1 memory cells were grouped together to identify Th1-type cells. Similarly, we combined the Tfh effector and Tfh memory clusters (Tfh type), as well as Tcmp and Tcm clusters (Tcm(p) type). For significantly differentially expressed genes, we calculated average expression profiles for the Th1, Tfh, and Tcm(p) types at individual timepoints using the *find.discriminant.genes* function of ProjecTILs (*Andreatta et al., 2021c*). The UCell algorithm (*Andreatta and Carmona, 2021b*) was applied on the CD4[+] T cell map subtypes to evaluate gene signatures for subtypes identified in a previous study (*Ciucci et al., 2019*). To exclude potential confounding factors, ribosomal-associated, sex-specific and TCR transcripts (*Magen et al., 2019*) were removed from signatures related to acute and chronic adaptation (*Supplementary file 3*).

## T cell clonal analysis

The CDR3 amino acid sequence for productive alpha-beta VDJ rearrangements obtained by scTCR-seq were used as unique 'barcodes' to identify individual T cell clones. The expansion level of a clone was calculated as the absolute number of cells with identical TCR sequence in a given sample, either in terms of CDR3 sequence or full nucleotide sequence. Expanded clones that were unique to a sample were denoted as private clones, expanded clones found in at least three samples were denoted as public clones. Merging and visualization of scTCR-seq data were performed using the scRepertoire package (*Borcherding et al., 2020*).

To measure gene expression- TCR relationship, we defined a 'clonotype bias' metric to quantify whether a given clonotype was preferentially composed of one of the T cell subtypes: $c = \max_i [ (f_i - q_i) / (1-q_i) ]$ where $f_i$ is the observed frequency for subtype i in the clonotype, and $q_i$ is the background frequency of subtype i in the whole sample. To assess statistical significance of measured clonotype bias scores, we generated N random permutations of the observed clonotype data, preserving clonal size and global subtype background frequencies. On the distribution of permuted clonotype bias scores, binned by clone size, we determined expected mean and standard deviation for each clone size bin. Z scores for observed clonotype bias scores were then calculated as the number of standard deviations from the background mean (Z score = 5 corresponds to a p-value $\sim 6*10^{-7}$). PWMs for the fate-biased TCR alpha CDR3 motifs identified by *Khatun et al., 2021* were kindly provided by the authors. We applied *glam2scan* (*Frith et al., 2008*) as in the original study to score these motifs on our data and rank clones based on individual motifs. We found that the motifs by Khatun et al. did not correlate with biased clones from our datasets (*Figure 4—figure supplement 1C*). Selecting clonotypes based on the 65% percentile of each of these motifs (as in the original study) did not enrich biased clones compared to their expected frequency. In particular, Tfh motifs were not predictive of Tfh bias and Th1 motifs were not predictive of Th1 bias (*Figure 4—figure supplement 1D*).

## Effect of cell cycling in defining reference spaces

To investigate the effect of cycling cells in the definition of low-dimensional embeddings, we re-analyzed the dataset by *Khatun et al., 2021* (LCMV acute day 10). Unsupervised analysis of two replicates from this study (replicates 1 and 2) showed that cycling cells clustered together and that cells in this cluster expressed a mixture of markers for Th1, Tfh, and Tregs (*Figure 5—figure supplement 1C-E*). Next, we asked whether projecting these cycling cells into our reference map allowed discriminating

the different subsets. Reference projection of cycling cells revealed that, while the majority of cells were assigned to Th1, more than 30% of cycling cells were predicted to be Tfh, Tcmp, and Treg (*Figure 5—figure supplement 1F*). The expression profile of a panel of marker genes for these cell subtypes corresponded closely with the reference profiles (with additional high expression of cycling markers for example *Mki67*, as these are all cycling cells), confirming that the cycling cluster was composed of a mixture of different cell types (*Figure 5—figure supplement 1G*). These analyses showed that cell cycling can mask differences between cell subtypes, and that failing to account for cell cycling signals leads to mixing of multiple cell types in low dimensional spaces.

## Effect of sequencing depth on reference-based annotation

Starting from the data generated in this study, as well as on an external dataset (*Khatun et al., 2021*), we applied the 'downsampleMatrix' function from the *scuttle* package (*McCarthy et al., 2017*) to generate downsampled scRNA-seq count matrices with increasingly lower sequencing depth (99%–10% of measured depth) for each sample in the study. These reduced-depth datasets were then systematically projected into the reference map, and we evaluated the agreement of the cell subtype annotation with the annotation of the full-depth dataset. We measured the classification agreement as a function of the minimal (1% quantile) or median (50% quantile) number of genes, and of the minimal and median number of UMIs. We observed that subtype classification was robust to sequencing depths down to 30% of the original sequencing depth, corresponding roughly to a median of 500 detected genes and around 1000 median UMIs per cell (*Figure 5—figure supplement 2B,E*). Moreover, disagreement in classification between full depth and downsampled depth was mostly affecting related cell subtypes, such as the effector and memory states of Tfh or Th1 cells, or Tcm cells and other memory subtypes (*Figure 5—figure supplement 2C,F*). On the whole, these experiments show that reference-based annotation is robust to sequencing depth for transcript counts and numbers of detected genes currently yielded by standard scRNA-seq technologies.

## Statistical analyses for flow cytometry data

Statistical significance was calculated with Prism software. Except where otherwise indicated in figure legends, error bars in graphs indicate standard deviation and statistical comparisons were done by one-way ANOVA test.

## Acknowledgements

We thank the NIH tetramer facility for reagents; the CCR and University of Rochester Flow Cytometry Core, the University of Rochester Genomics Research Center and the NIH High performance computing cluster for assistance; D McGavern, T Mosmann and F David for technical assistance; R Bosselut for supporting the research; D Goldstein, M Malik and the NCI Office of Science and Technology Resources for their support. We would also like to thank Prof. Carolyn King and David Schreiner at the University of Basel for critical reading of the manuscript. This work was supported by the University of Rochester, and Intramural Research Program of the National Cancer Institute, Center for Cancer Research (CCR), National Institutes of Health, and by the Swiss National Science Foundation (SNF project 180010). The CCR Single Cell Analysis Facility is funded by the Frederick National Laboratory for Cancer Research, Contract HHSN261200800001E. Sequencing was performed with the CCR Genomics Core and the University of Rochester Genomics Research Center.

## Additional information

### Funding

| Funder | Grant reference number | Author |
| --- | --- | --- |
| Schweizerischer Nationalfonds zur Förderung der Wissenschaftlichen Forschung | PZ00P3_180010 | Santiago J Carmona |

| Funder | Grant reference number | Author |
|---|---|---|

The funders had no role in study design, data collection and interpretation, or the decision to submit the work for publication.

## Author contributions

Massimo Andreatta, Conceptualization, Resources, Data curation, Software, Formal analysis, Validation, Investigation, Visualization, Methodology, Writing – original draft, Writing – review and editing; Ariel Tjitropranoto, Zachary Sherman, Validation, Methodology; Michael C Kelly, Resources, Methodology, Project administration; Thomas Ciucci, Santiago J Carmona, Conceptualization, Resources, Data curation, Software, Formal analysis, Supervision, Funding acquisition, Validation, Investigation, Visualization, Methodology, Writing – original draft, Project administration, Writing – review and editing

## Author ORCIDs

Massimo Andreatta http://orcid.org/0000-0002-8036-2647
Ariel Tjitropranoto http://orcid.org/0000-0001-5525-5236
Michael C Kelly http://orcid.org/0000-0003-0654-2778
Thomas Ciucci http://orcid.org/0000-0002-5828-0207
Santiago J Carmona http://orcid.org/0000-0002-2495-0671

## Ethics

This study was performed under the protocol UCAR 2020-003 approved by the University of Rochester Committee on Animal Resources.

## Decision letter and Author response

Decision letter https://doi.org/10.7554/eLife.76339.sa1
Author response https://doi.org/10.7554/eLife.76339.sa2

---

# Additional files

## Supplementary files

• Transparent reporting form
• Supplementary file 1. Summary of scRNA-seq and scTCR-seq data generated in this study.
• Supplementary file 2. Gene signatures for CD4$^+$ T cell subtypes.
• Supplementary file 3. Auxiliary gene sets used in this study.

## Data availability

Sequence data are deposited in the NCBI Gene Expression Omnibus under accession numbers GSE182320 and GSE200635. The new reference atlas can be downloaded (DOI: 10.6084/m9.figshare.16592693) or accessed via the web portal (https://spica.unil.ch/refs/viral-CD4-T). All code sources are available at https://github.com/carmonalab/ProjecTILs (copy archieved at swh:1:rev:b8b-b396674697a3e6ca53967ca768f2e2fb7e61c) and https://github.com/carmonalab/ProjecTILs_CaseStudies.

The following datasets were generated:

| Author(s) | Year | Dataset title | Dataset URL | Database and Identifier |
|---|---|---|---|---|
| Ciucci T, Carmona S | 2021 | Single-cell gene expression of virus-specific CD4 T cells in response to acute and chronic infection | https://www.ncbi.nlm.nih.gov/geo/query/acc.cgi?acc=GSE182320 | NCBI Gene Expression Omnibus, GSE182320 |
| Ciucci T, Carmona S | 2022 | Single-cell gene expression of tumor-specific CD4 T cells | https://www.ncbi.nlm.nih.gov/geo/query/acc.cgi?acc=GSE200635 | NCBI Gene Expression Omnibus, GSE200635 |

The following previously published datasets were used:

| Author(s) | Year | Dataset title | Dataset URL | Database and Identifier |
|---|---|---|---|---|
| Cui W, Khatun A | 2020 | Single-cell lineage mapping of a diverse virus-specific naïve CD4 T cell repertoire | https://www.ncbi.nlm.nih.gov/geo/query/acc.cgi?acc=GSE158896 | NCBI Gene Expression Omnibus, GSE158896 |
| Kuenzli M, Schreiner D, Roux J, King C | 2019 | Single-cell RNA-sequencing of spleen memory CD4+ T cells | https://www.ncbi.nlm.nih.gov/geo/query/acc.cgi?acc=GSE134157 | NCBI Gene Expression Omnibus, GSE134157 |
| Bosselut R, Ciucci T | 2018 | Single-cell gene expression of anti-viral WT and Thpok-deficient effector and memory T cells | https://www.ncbi.nlm.nih.gov/geo/query/acc.cgi?acc=GSE121002 | NCBI Gene Expression Omnibus, GSE121002 |

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
