## [Editor Report]

This paper uses single-cell genomics to examine the heterogeneity of virus-specific CD4 T cells over time in both acute and chronic viral infection. Further, the authors build a comprehensive atlas of the transcriptional evolution of virus-specific CD4 T cell responses that could be used as a reference tool to interpret other datasets. This work characterizes how the antiviral CD4 T cell transcriptional landscape changes with time and will be of broad interest to those that study acute and chronic CD4 T cell responses.

---

## [Decision Letter]

**Decision letter after peer review:**

Thank you for submitting your article "A CD4^+^ T cell reference atlas delineates subtype-specific adaptation during acute and chronic viral infections" for consideration by *eLife*. Your article has been reviewed by 2 peer reviewers, and the evaluation has been overseen by a Reviewing Editor and Tadatsugu Taniguchi as the Senior Editor. The following individual involved in review of your submission has agreed to reveal their identity: Laura M Snell (Reviewer #1).

Essential revisions:

1) The reviewers agreed that additional details, clarity, validation, and broader integration of the proposed atlas would strengthen the conclusions and usefulness of the study. However, if these additional analyses lead to altered interpretation or utility of the atlas then the authors should revise the work accordingly.

2) Please pay close attention to the detailed recommendations provided by Rev #2.

*Reviewer #1 (Recommendations for the authors):*

1) Viral titers should be paired to the different antiviral CD4 T cell transcriptional outcomes. Although LCMV Armstrong is quickly cleared, the clearance kinetics of LCMV Clone 13 can vary dramatically from laboratory to laboratory. The paper simply says that LCMV Clone 13 persists, but it is important to demonstrate how close the virus is to being cleared at the late chronic timepoint, as viral titer will impact transcriptional phenotypes. It will be relevant to have this information when using the reference atlas on other LCMV datasets which may clear with slightly altered kinetics.

2) Gene expression in Th1, Tfh and Tcm(p) is well characterized across time. Does gene expression in the Th1 memory, Tfh memory and Tcm change across early memory to late memory during acute infection? How does gene expression in the Tfh memory in late chronic infection compare to that of early memory (day 21 in both models)?

3) While the changes in proportions of the cell clusters across time and in acute versus chronic viral infection are demonstrated, a big contraction of virus-specific CD4 T cell numbers would be expected between Day 7 and Day 21 in both acute and chronic infection. As such, it would be relevant to also show the absolute number of cells in each cluster across the timepoints to get an accurate depiction of whether the enhancement in proportions of memory clusters also translated to an enhancement in absolute numbers of these clusters, or whether the numbers of memory cells in each cluster are simply maintained across the timepoints.

4) The text says that the atlas with the reference projection algorithm can enable interpretation of CD4 states across models, although all the examples given were based on LCMV datasets. Can the reference atlas accurately determine Th1/Tfh phenotypes from non-LCMV CD4 datasets? Many other models also drive Th1/Tfh differentiation. Single-cell analysis has been done on the discrimination of Th1/Tfh in malaria for instance: Lonnberg T et al. Sci Immunol. 2017 etc. and new data is emerging characterizing CD4s in various cancer models. Does the reference atlas hold up when determining CD4 subsets from data that is not LCMV-based?

5) The figure legends could benefit from more detail. In figure 1 for instance it is unclear if the UMAPs are based on a representative sample or the merged data of all samples. Also, the tissue of origin where the cells were sorted from should be mentioned for the reader's clarity.

*Reviewer #2 (Recommendations for the authors):*

1) The sequencing batches used to construct the 'atlas' contain biologically distinct samples (Figure 1A-B). Therefore, prior to integration, both technical and biological differences will drive cell separation. In such instances it is useful to have at least one cell population present in all batches to verify integration performance – cells from equivalent populations should produce a joint overlapping cluster whereas biologically distinct populations such as central memory T cells and exhausted T cells should produce distinct clusters. By difficult to understand experimental design, this paper does not seem to have any such populations so the performance of the integration is difficult to assess. Even so, the authors could quantify the degree of alignment between clusters in the d21 Clone 13 samples present in batches 2 and 3, and the d7 Arm samples present in Batch 1 and 2. Based on Figure S2A which is the only data related to integration performance, there is significant heterogeneity between biological replicates. For example, Tregs are virtually absent from the second Late Chronic biological replicate whereas the 'Tfh memory' subset is highly abundant compared to the first replicate. Similarly, the cluster frequencies of the low-frequency clusters look very different between replicates in the Early Memory (d21 Arm) group. Given this uncertainty about integration performance, it is difficult to interpret the subsequent data as it could be partially explained by technical variation between batches.

2) The TCR analysis does not address prior work by Khatun et al. (JEM 2020) which showed that the Tfh bias of certain TCR sequences could be predicted in independent mice. The authors' analysis is limited to stating the degree of bias in each clonotype frequency group. Did the authors attempt to replicate the observation by Khatun et al.? What was the overlap between CDR3 motifs? What was the overlap in motifs between Khatun et al. and this study?

3) The 'atlas' functionality is limited to a superficial demonstration of projecting several LCMV CD4 T cell dataset onto the authors' dataset. There is no data quantifying the performance of this integration in absolute terms or relative to other methods. For example, given that the 9 clusters defined by the authors are previously known CD4 T cell subsets, what is the advantage of using this method compared to quantifying the expression of existing marker gene sets in the primary datasets? What is the performance of this method compared to manual integration of individual datasets?

4) What is the effect of sequencing depth on integration performance? Would low-depth datasets produce annotation results with the most central clusters dominant due to lack of specific, cluster-defining lowly expressed genes? What is the minimum depth at which technical effects would not drive integration? This type of information is essential if the 'atlas' is to be used as a tool, otherwise the resulting misannotations could do more harm than good to the users.

5) It is unclear how the removal of cell cycle genes from the initial dataset affects interpretation and integration. Given that cell cycle state and cell fate are causally linked in T cells, would the removal of cell cycle genes not obscure some meaningful transcriptomic differences between populations? Are the cell cycle genes in dividing effector cells the same as in dividing early memory cells?

6) The experimental validation of this dataset is limited to showing that CD4 T cells in persistent infection express more EOMES than T cells in acutely infected mice and that they express lower levels of THPOK. However, what is the global alignment between flow cytometry data presented in Figure 1 and the scRNAseq data? Were any of the cluster frequencies predicted by the scRNAseq data validated using a protein panel?

[Editors' note: further revisions were suggested prior to acceptance, as described below.]

Thank you for resubmitting your work entitled "A CD4^+^ T cell reference map delineates subtype-specific adaptation during acute and chronic viral infections" for further consideration by *eLife*. Your revised article has been evaluated by Tadatsugu Taniguchi (Senior Editor) and a Reviewing Editor.

The manuscript has been improved but there is one remaining issue that needs to be addressed, as outlined below:

*Reviewer #1 (Recommendations for the authors):*

In general this reviewer is satisfied with the revisions to the manuscript, however, one point needs to be better clarified. When the tumor-specific CD4 T cells from TILs were projected into the reference map 40-50% of them mapped into Th1 effectors. Yet upon further analysis and reclustering, these cells ended up being a completely distinct population of cells from the viral effector Th1. Thus, this reviewer is worried this could lead to misinterpretation and incorrect identification of subsets when using the reference map on other systems with unrepresented subsets not in the reference map. Could the authors comment on/clarify this point? It would be helpful to discuss the additional steps needed to verify that the corresponding states determined from projecting one's data into the reference map have similar gene profiles, and if they do not, how to address and identify these novel populations not represented in the map.

*Reviewer #2 (Recommendations for the authors):*

The authors have addressed my concerns. I can now recommend publication.

---

## [Author Response]

Essential revisions:1) The reviewers agreed that additional details, clarity, validation, and broader integration of the proposed atlas would strengthen the conclusions and usefulness of the study. However, if these additional analyses lead to altered interpretation or utility of the atlas then the authors should revise the work accordingly.2) Please pay close attention to the detailed recommendations provided by Rev #2.

We thank reviewers and editors for their interest in the manuscript and especially for their valuable feedback. We have addressed all the reviewers' comments, supported by new data and several new analyses. In particular, we demonstrate the robustness and generalizability of our reference map and projection method by analyzing data from multiple tissues and viral infection models, as well as following genetic and therapeutic perturbations. In addition, we generated and analyzed a novel dataset derived from tumor-specific CD4^+^ T cells, showing the utility of our framework to interpret cell diversity even beyond the cell states currently present in the reference map.

The new results are shown in Figure 5 (new panels D and E), in two new main Figures 6 and 7, in new panels in supplemental figures Figure 1—figure supplement 1, Figure 2—figure supplement 1, Figure 4—figure supplement 1, and new supplemental figures Figure 5—figure supplement 1, Figure 5—figure supplement 2, and Figure 5—figure supplement 3. Please note that names of the supplemental figures have been changed and associated to main figures per request of the journal editors.

The manuscript has been modified to accommodate the new data and address the comments by the referees.

Reviewer #1 (Recommendations for the authors):1) Viral titers should be paired to the different antiviral CD4 T cell transcriptional outcomes. Although LCMV Armstrong is quickly cleared, the clearance kinetics of LCMV Clone 13 can vary dramatically from laboratory to laboratory. The paper simply says that LCMV Clone 13 persists, but it is important to demonstrate how close the virus is to being cleared at the late chronic timepoint, as viral titer will impact transcriptional phenotypes. It will be relevant to have this information when using the reference atlas on other LCMV datasets which may clear with slightly altered kinetics.

This is an important technical point since viral replication can vary at later timepoints, especially in the presence of CD4^+^ T cells. We agree with the reviewer that, ideally, viral titers could be determined before proceeding to the single-cell capture and RT-PCR. However, our experience with single-cell transcriptomic technology showed that optimal capture, *i.e.* cell number and gene coverage, sharply decreases ~4h after animals are sacrificed. Therefore, determining viral titers, even by qPCR, represents a logistical challenge that could have technically jeopardized the study or compromised data quality.

However, we performed a phenotypic characterization of T cell populations by flow cytometry for each capture, in particular focusing on the persistence of the “exhaustion” marker PD1 and the diminution of IL7R expression by the virus-specific CD4^+^ T cell populations. As shown in Author response image 1, we confirmed that, for both “Late chronic” captures, CD4^+^ T cells were PD1^high^ IL7R^intermadiate/low^ unlike their “Early Memory” counterparts which were PD1^low^ IL7R^high^.

**Author response image 1. sa2fig1:** Flow cytometry expression of PD1 and IL7R on naïve and GP66 CD4^+^ T cells for each of the indicated sample processed for scRNAseq capture.

Moreover, to put our results in the context of known LCMV Cl13 viral titers, we compared our transcriptional profiles to those of a recent study of LCMV-specific CD4^+^ T cells at day 33 of chronic infection, in which plasma viral titers have been determined (in the order of 10^5^ PFU per ml of plasma) (Snell *et al.*, Nature Immunology 2021, PMID: 34795443). Although this study used (transgenic-TCR) SMARTA cells and our data consist of polyclonal GP66-specific tetramer sorted cells, projection of the data by Snell *et al.* into our reference map showed a remarkable similarity to our Late Chronic samples, both in terms of subtype composition and their transcriptional profiles. Our automated analysis confirmed that, compared to isotype control, anti-PD-L1 treatment induced amplification of Th1 SMARTA cells, and that these upregulated Th1-associated genes including *Klrd1*, *Ly6c2*, *Ctla2a, Plac8* and *Lgals3*, in agreement with the findings by Snell *et al.* These results are shown in the new panel E of Figure 5 and new Figure 5—figure supplement 3:

This analysis is described in the manuscript:

“Next, we aimed at using our reference map to interpret the effect of immunotherapies. To this end, we projected scRNA-seq data of virus-specific CD4^+^ T cells isolated from mice with LMCV chronic infection, after treatment with an anti-PD-L1 antibody (Snell et al. 2021). While control samples showed a similar subtype distribution to our late chronic samples (Figure 3C, Figure 5E), anti-PD-L1 treatment increased the relative proportion of Th1 effectors (Figure 5E). Expression profiles for all major subtypes in this dataset largely matched those of our reference (Figure 5—figure supplement 3A), including the expression of exhaustion markers that was similar to that of the chronic infection samples of the reference map (Figure 5—figure supplement 3B). Notably, Th1 effector cells after anti-PD-L1 treatment, upregulated a Th1-associated gene module that includes *Klrd1*, *Plac8*, *Ctla2a,* and *Ly6c2* (Figure 5—figure supplement 3C), confirming the findings by Snell et al. 2021.”

These two independent validations demonstrate that our late chronic datasets were isolated at a timepoint when CD4^+^ T cells were actively responding to viral antigens.

2) Gene expression in Th1, Tfh and Tcm(p) is well characterized across time. Does gene expression in the Th1 memory, Tfh memory and Tcm change across early memory to late memory during acute infection? How does gene expression in the Tfh memory in late chronic infection compare to that of early memory (day 21 in both models)?

This is an interesting point that we briefly discussed for Tfh memory (lines 174 to 180), since we observed that this subtype undergoes substantial changes in late memory compared to the early memory phase. These changes include the downregulation of Tfh effector program (including *Icos* and *Il21*) and are accompanied by the upregulation of memory genes like *Il7r*. We did note a modest increase in “memory-associated” genes for Th1 and Tcm(p) subtypes from early to late memory, though not to the same extent as Tfh memory cells (Figure 3B). Additional comparisons during Acute timepoints and subsets are included in Supplementary File 2 – Acute Adaptation.

Concerning the differences between Acute day 21 (Early Memory) to Chronic day 21 (Late Chronic), we now included these comparisons in Supplementary File 2 – Chronic Adaptation. As expected, many of the genes upregulated in Late Chronic are associated with T cell activation and T cell dysfunction in response to persistent TCR signaling and response to viral antigens. In particular, Tfh cells in late chronic infection upregulated inhibitory receptors such as *Lag3*, *Pdcd1* and *Ctla4*, and transcription factors associated with exhaustion such as *Tox* and *Bhlhe40* (Supplementary File 2 – Chronic Adaptation).

3) While the changes in proportions of the cell clusters across time and in acute versus chronic viral infection are demonstrated, a big contraction of virus-specific CD4 T cell numbers would be expected between Day 7 and Day 21 in both acute and chronic infection. As such, it would be relevant to also show the absolute number of cells in each cluster across the timepoints to get an accurate depiction of whether the enhancement in proportions of memory clusters also translated to an enhancement in absolute numbers of these clusters, or whether the numbers of memory cells in each cluster are simply maintained across the timepoints.

We agree that including absolute T cell numbers will provide a clearer view of the immune response. To address this point, we estimated the absolute number of T cells for all subtypes at early and late timepoints by multiplying total cell numbers in spleen determined by flow-cytometry across multiple experiments (new panel Figure 2—figure supplement 1F) and cluster proportions (shown in Figure 3A,C and Figure 2—figure supplement 1D). We observed that for both types of infection the number of memory cells at later timepoints exceed the number of the same cells at day 7 (new panel Figure 2—figure supplement 1G). Since there is no evidence that these memory subsets are actively proliferating at early timepoints (see reply to Reviewer #2), one possibility is that at least part of the memory pool is derived from effector subsets generated at earlier timepoints. This is consistent with previous studies showing that effector subtypes can transition – although to a lower extent than “memory precursors” – to memory subtypes (Harrington *et al.*, Nature 2008, PMID: 18322463; Marshall *et al.*, Immunity 2011; PMID: 22018471). We integrated these results in the Figure 2—figure supplement 1F-G and lines 163-164.

4) The text says that the atlas with the reference projection algorithm can enable interpretation of CD4 states across models, although all the examples given were based on LCMV datasets. Can the reference atlas accurately determine Th1/Tfh phenotypes from non-LCMV CD4 datasets? Many other models also drive Th1/Tfh differentiation. Single-cell analysis has been done on the discrimination of Th1/Tfh in malaria for instance: Lonnberg T et al. Sci Immunol. 2017 etc. and new data is emerging characterizing CD4s in various cancer models. Does the reference atlas hold up when determining CD4 subsets from data that is not LCMV-based?

This is a great suggestion. Indeed, our method can be used to annotate and interpret datasets from other models and tissues. To illustrate this, we analyzed T cells isolated from different tissues and infection models. In particular, our method accurately interpreted data from LCMV-specific CD4^+^ T cells in the liver (Künzli et al., Science Immunology, PMID: 32144185) and from influenza infection in the lung and lymph nodes (Swarnalekha et al., Science Immunology 2021, PMID: 33419790). These results are presented in the new Figure 6 and discussed in a new section of the manuscript: “Diversity of virus-specific CD4^+^ T cells across tissues”.

Briefly, we recapitulate the findings by Swarnalekha et al. that (*a)* Th1 subtypes dominate in lung populations at early time points, unlike their counterpart in the lymph nodes after influenza infection, (*b)* lung CD4^+^ T cell populations become enriched in Tfh-like states over time, (*c)* cells in the lungs display a tissue-resident gene module expressed across Tfh memory, Tfh effector and Th1 effector subtypes (Figure 6A-D). Interestingly, this gene module was also found to be significantly higher in liver compared to spleen in the context of LCMV infection (Figure 6F), suggesting a tissue residency program that is partly conserved across infections and cell states, and highlighting the power of our framework for discovering such programs.

In addition, we included a new analysis showing that our tool is useful to interpret the heterogeneity of CD4^+^ T cells beyond viral infections. These additional analyses are presented in the new Figure 7 and discussed in a new section of the manuscript: “Reference map projection to explore CD4^+^ T cell diversity beyond viral infections”.

Briefly, we generated an entirely new scRNA-seq dataset derived from tumor-specific CD4^+^ T cells (deposited in GEO under identifier GSE200635). Projection of these data into our viral infection-derived reference map revealed that tumor-specific CD4^+^ T cell populations in the tumor-draining lymph nodes were dominated by Tfh cells while tumors were enriched in Tregs, as well as by a distinct non-Treg, non-Tfh state (Figure 7A-B). Further investigation showed that, compared to Th1 effectors in response to viruses, these effector cells in the tumor differentially expressed Th2-associated genes (e.g. *Ccr8*, *Tgfb1*, *Ccl1*, *Il5*, *Il13*, *Igfbp7)* (Figure 7E-F).

5) The figure legends could benefit from more detail. In figure 1 for instance it is unclear if the UMAPs are based on a representative sample or the merged data of all samples. Also, the tissue of origin where the cells were sorted from should be mentioned for the reader's clarity.

More details have been added to clarify the figure legends throughout the manuscript.

Reviewer #2 (Recommendations for the authors):1) The sequencing batches used to construct the 'atlas' contain biologically distinct samples (Figure 1A-B). Therefore, prior to integration, both technical and biological differences will drive cell separation. In such instances it is useful to have at least one cell population present in all batches to verify integration performance – cells from equivalent populations should produce a joint overlapping cluster whereas biologically distinct populations such as central memory T cells and exhausted T cells should produce distinct clusters. By difficult to understand experimental design, this paper does not seem to have any such populations so the performance of the integration is difficult to assess. Even so, the authors could quantify the degree of alignment between clusters in the d21 Clone 13 samples present in batches 2 and 3, and the d7 Arm samples present in Batch 1 and 2. Based on Figure S2A which is the only data related to integration performance, there is significant heterogeneity between biological replicates. For example, Tregs are virtually absent from the second Late Chronic biological replicate whereas the 'Tfh memory' subset is highly abundant compared to the first replicate. Similarly, the cluster frequencies of the low-frequency clusters look very different between replicates in the Early Memory (d21 Arm) group. Given this uncertainty about integration performance, it is difficult to interpret the subsequent data as it could be partially explained by technical variation between batches.

We thank the reviewer for these comments, which allow us to clarify a few points about batch effects, biological variability, and the power of our approach.

In their comment, we believe the reviewer refers to Figure 2A-B and not Figure 1, since Figure 1A-B are not for sequencing data but from spectral flow cytometry, for which the cells from all 4 conditions were analyzed at the same time. For additional clarification, the latter consisted in infecting animals at different timepoints so that the cell processing and analyses will be performed on the same day. This represents, for each experiment, 16-20 mice for which splenocytes were isolated, stained and acquired simultaneously to limit batch effect.

Regarding the integration of scRNA-seq data (Figure 2 A-B), we agree with the reviewer that data presented in Figure 2—figure supplement 1A were not sufficient to convincingly assess integration quality and reproducibly of the many replicates, which we have now addressed extensively (see below).

The reviewer commented that: “In such instances it is useful to have at least one cell population present in all batches to verify integration performance – cells from equivalent populations should produce a joint overlapping cluster whereas biologically distinct populations such as central memory T cells and exhausted T cells should produce distinct clusters.”. While we agree that it can be useful to have one cell population present in all batches, we have previously shown (Andreatta and Carmona (2021) PMID:32845323) that our integration method STACAS does not require that every batch shares a population with every other batch. Instead, it is sufficient that every batch shares one cell population with *at least* another batch to enable generating an integration guide tree. This is indeed the case, now highlighted in a new table (Supplementary File 1 B).

For instance, as the reviewer mentioned, cell populations present in d21 in Clone 13 and Arm samples (Early Memory and Late Chronic) are present in both batches 2 and 3, and cell populations of d7 Armstrong samples were present in both Batch 1 and 2; allowing for a pairwise batch integration strategy (e.g. B2-B3; (B2-B3)-B1). We acknowledge that this information was not clear in the first version, which was important to understand the experimental design.

Regarding heterogeneity between biological replicates (“Based on Figure S2A which is the only data related to integration performance, there is significant heterogeneity between biological replicates.”), we provide evidence that variability between biological replicates is within an expected range (by flow cytometry as well as by re-analysis of data by Snell et al. Nature Immunology 2021), and much lower than variability between conditions. Importantly, unlike the vast majority of scRNA-seq studies, we provide at least two biological replicates for every condition, allowing us to assess similarity between replicates (as shown in Figure 2—figure supplement 1A).

To better illustrate the proportion of each cell population across biological replicates, we show in the new Figure 2—figure supplement 1D-E the strong similarities in the distribution of all cell states across samples. In particular, despite minimal variations in the percentages of each state, all replicates from each condition are composed of similar subtype composition. Critically, samples clustered by condition rather than by batch (i.e. biological replicates bear the largest similarity to each other, despite batch effects). Validating the proportions of these populations by spectral cytometry showed that they are comparable to the subtypes defined from scRNA-seq (new Figure 1—figure supplement 1 E). These results are presented as new Figure 2—figure supplement 1D-E and discussed lines 129-132:

“All subtypes were present in similar proportions across biological replicates, and samples clustered by condition rather than by batch, further confirming a successful data integration (Figure 2—figure supplement 1D-E). Similar subtype proportions were confirmed by spectral cytometry (Figure 1—figure supplement 1E)”

Regarding the 3 minor states, the IFNI-stimulated population was mostly found in chronic samples, where it consistently represented 6-10% of the cells, and the Eomes^HI^ state was mostly seen in the two late chronic replicates, with a frequency of 3-4%. Finally, Tregs represented a very small fraction (consistently below 1% in all samples); preventing us to make any robust conclusion concerning their proportions variability.

As the reviewer correctly points out, we observed indeed that the Tfh_memory subtype was relatively more variable between Late Memory replicates (i.e. 24% vs 10%; ~17% on average). However, these proportions are (a) very similar to the Tfh_memory percentage confirmed by flow cytometry (~15%, Figure 1—figure supplement 1E), and (b) very similar to the Tfh_memory proportion determined in an independent scRNA-seq dataset of chronic infection (Figure 5E; ~15% Tfh memory in the untreated mice).

With all that said, precisely determining the degree of biological variability in the proportion of Tfh_memory is out of the scope of this manuscript, and these variabilities do not affect the conclusions of our study, given that gene expression profiles in each subtype were consistent across replicates.

2) The TCR analysis does not address prior work by Khatun et al. (JEM 2020) which showed that the Tfh bias of certain TCR sequences could be predicted in independent mice. The authors' analysis is limited to stating the degree of bias in each clonotype frequency group. Did the authors attempt to replicate the observation by Khatun et al.? What was the overlap between CDR3 motifs? What was the overlap in motifs between Khatun et al. and this study?

In line with our results, the study by Khatun *et al.* found that while the majority of clonotypes were not biased towards a specific fate, a subset of clonotypes were seen to preferentially develop into one particular lineage. In addition, they found that certain TCR α chains, but not β chains, were enriched in Tfh cells compared to other fates. Based on this observation, they derived CDR3 motifs of the TCR α chains that distinguished biased clones in their dataset, obtaining 27 (11 Th1-specific and 16 Tfh-specific) motifs. When tested on left-out mice not used to compute the motifs, they found that no Th1 motifs were predictive of Th1 bias, 2 Tfh motifs were predictive of Tfh bias, and one Tfh motif showed in fact significant preference towards the opposite lineage. The remaining 24 “fate-biased” motifs were not reproduced in the independent mice. We believe this is very weak evidence for the predictability of fate based on TCR sequence.

Nevertheless, we evaluated the PWMs for the motifs described by Khatun et al. (kindly provided by the authors) on our data to determine whether they were predictive of clonotype fate. As shown in the new Figure 4—figure supplement 1C, motif scores for the 7 motifs from Figure 4 of Khatun et al. (calculated using *glam2scan* [Frith et al. (2008)] as in the original paper) do not correlate with biased clones from our datasets. Selecting clonotypes based on the 65% percentile of each of these motifs (threshold used by Khatun et al) does not enrich biased clones compared to their expected frequency (column “All”). In particular, Tfh motifs are not predictive of Tfh bias and Th1 motifs are not predictive of Th1 bias (Figure 4—figure supplement 1D). These analyses were included as part of Figure 4—figure supplement 1 (panels C and D), and discussed on lines 281-283 and 710-717.

“We did not observe any robust CDR3 motif associated with biased clonotypes, and previously reported fate-biased CDR3 motifs (Khatun et al., 2021) were not predictive of clonotype lineage on our data (Figure 4—figure supplement 1C-D).”

“PWMs for the fate-biased TCR α CDR3 motifs identified by Khatun et al. (2021) were kindly provided by the authors. We applied glam2scan [PMID:18437229] as in the original study to score these motifs on our data and rank clones based on individual motifs. We found that the motifs by Khatun et al. did not correlate with biased clones from our datasets (Figure 4—figure supplement 1C). Selecting clonotypes based on the 65% percentile of each of these motifs (as in the original study) did not enrich biased clones compared to their expected frequency. In particular, Tfh motifs were not predictive of Tfh bias and Th1 motifs were not predictive of Th1 bias (Figure 4—figure supplement 1D).”

3) The 'atlas' functionality is limited to a superficial demonstration of projecting several LCMV CD4 T cell dataset onto the authors' dataset. There is no data quantifying the performance of this integration in absolute terms or relative to other methods. For example, given that the 9 clusters defined by the authors are previously known CD4 T cell subsets, what is the advantage of using this method compared to quantifying the expression of existing marker gene sets in the primary datasets? What is the performance of this method compared to manual integration of individual datasets?

Given the lack of robust methods or validated gene signatures for CD4^+^ T cell subtype classification, the only alternative is, as the reviewer suggests, manual annotation. Since there is no standardized way of performing a manual annotation, this process is usually both highly time-demanding and subjective. As an increasingly large body of scRNA-seq data becomes available, unsupervised analysis of individual datasets will become untenable. An automated and scalable system provides a unified framework to bring multiple datasets into the same space, to assign consistent labels, and to perform meta-analyses with robust criteria. We have previously demonstrated the usefulness and accuracy of such an approach in other contexts, and compared its performance with alternative methods (Andreatta *et al.* Nature Communications 2021).

In this revised version, we perform several additional analyses showing that our tool accurately predicts CD4^+^ T cell subtype composition changes across tissues in two different viral infections (LCMV infection spleen vs. liver; and flu in lymph node vs. lung; new Figure 6), upon anti-PD-L1 therapy (new Figure 5E, new Figure 5—figure supplement 3), and in the context of cancer (new Figure 7). Please see the detailed answer to reviewer #1 for these analyses. Using a fully automated method, we were able to recapitulate non trivial conclusions from previous studies, including (a) a large proportion of Tfh-like cells in lungs at late timepoints after influenza infection, (b) expression of a conserved non-lymphoid tissue residency gene module that was predictive across infection types, (c) amplification of Th1 cells following anti-PD-L1 therapy; showing that this reference-based analysis framework can save huge efforts in data analysis and subjectivity of manual annotations. Finally, we showed that even such an ‘incomplete’ CD4^+^ T cell map (built from viral infection data) provides a scaffold to identify novel states, such as that of tumor-specific CD4^+^ T cells (new Figure 7).

To address the reviewer’s concern about a potential misunderstanding of the word ‘atlas’ (i.e. collection of maps) in this context, we have replaced it with ‘map’ throughout the manuscript.

4) What is the effect of sequencing depth on integration performance? Would low-depth datasets produce annotation results with the most central clusters dominant due to lack of specific, cluster-defining lowly expressed genes? What is the minimum depth at which technical effects would not drive integration? This type of information is essential if the 'atlas' is to be used as a tool, otherwise the resulting misannotations could do more harm than good to the users.

We agree this is an important technical point that was not sufficiently addressed. We performed new analyses to assess the effect of sequencing depth on reference projection accuracy. Starting from the data generated in this study, as well as on an external dataset (Khatun et al. 2021), we applied the ‘downsampleMatrix’ function from the *scuttle* package (McCarthy et al. 2017) to generate downsampled scRNA-seq count matrices with increasingly lower sequencing depth (99% to 10% of measured depth) for each sample in the study. These reduced-depth datasets were then systematically projected into the reference map, and we evaluated the agreement of the cell subtype annotation with the annotation of the full-depth dataset (Figure 5—figure supplement 2A,D). We were also able to evaluate how the classification agreement was maintained as a function of the minimal (1% quantile) or median (50% quantile) number of genes, and of the minimal and median number of UMIs (Figure 5—figure supplement 2B,E). We observed that subtype classification was robust to sequencing depths down to 30% of the original sequencing depth, corresponding roughly to a median of 500 detected genes and around 1000 median UMIs per cell. Moreover, disagreement in classification between full depth and downsampled depth was mostly affecting related cell subtypes, such as the effector and memory states of Tfh or Th1 cells, or Tcm(p) cells and other memory subtypes (results for downsampling to 30% of select samples Figure 5—figure supplement 2 C,F). On the whole, these experiments show that reference-based annotation is robust to sequencing depth for transcript counts and numbers of detected genes currently yielded by standard scRNA-seq technologies. Thus, users can confidently use our tool without significant risks of misannotation. These results are presented in new Figure 5—figure supplement 2 and discussed in the new section: “Effect of sequencing depth on reference-based annotation” (lines 734-749).

5) It is unclear how the removal of cell cycle genes from the initial dataset affects interpretation and integration. Given that cell cycle state and cell fate are causally linked in T cells, would the removal of cell cycle genes not obscure some meaningful transcriptomic differences between populations? Are the cell cycle genes in dividing effector cells the same as in dividing early memory cells?

We thank the reviewer for allowing us to clarify this point. First, we would like to point out that cell cycle genes are not removed from the datasets. The misunderstanding might stem from the fact that, as part of our data integration procedure, cell cycle genes (as well as TCR genes and others) are excluded from variable gene selection (this has been clarified in line 651-653). This procedure is done to mitigate the effect of cell cycling, which causes dramatic transcriptional changes irrespective of cell subtype or basal transcriptional programs. Instead, we aim at clustering cells based on functionally distinct subtypes (i.e. the subtype-defining core transcriptional programs) rather than based on proliferation status, or other more transient signals. In practice, this only leads to excluding a limited number of genes that fell within the set of most highly variable genes (800 for this map). Of note, the exclusion of cell cycle genes from the variable features has much less impact than the typical ‘cell cycle regression’ procedure used to mitigate cell cycle, which affects all genes measurements (reviewed in Lueken et al. (2019) Mol Syst Biol).

To address more directly the reviewer’s concern about the possibility that we might be missing important biological information encoded in cell cycle genes with our procedure, we re-analyzed the dataset by Khatun et al. (LCMV acute day 10) (Figure 5—figure supplement 1C-G). Unsupervised analysis shows that cycling cells cluster together (Figure 5—figure supplement 1C-D), and that cells in this cluster express a mixture of markers for Th1 and Tfh (Figure 5—figure supplement 1D-E). A fraction of cells in the cycling cluster also express high levels of *Foxp3*, a marker of Tregs (Figure 5—figure supplement 1E). Thus, unsupervised clustering practically groups cells solely based on their proliferation status, regardless of their functional phenotype.

Next, we asked whether projecting these cycling cells into our reference map allows discriminating the different subsets. Projection of all cells from the cycling cluster reveals that, while the majority of cells were assigned to Th1, more than 30% of cycling cells were predicted to be Tfh, Tcmp and Treg (Figure 5—figure supplement 1F). The expression profile of a panel of marker genes for these cell subtypes corresponded closely with the reference profiles (with additional high expression of cycling markers e.g. *Mki67*, as these are all cycling cells), confirming that the cycling cluster was composed of a mixture of different cell types (Figure 5—figure supplement 1G).

These results show that cell cycling can indeed mask differences between cell subtypes, and that failing to account for cell cycling signals leads to mixing of multiple cell types in low dimensional spaces; thus validating our choices. Cell cycling, which is an essential characteristic to consider in a biological system, can then be evaluated as a separate quantity that is orthogonal to cell subtypes, and cycling cells can be annotated by projection into the reference space.

These results are presented in new Figure 5—figure supplement 1 and described in section: “Effect of cell cycling in defining reference spaces” (lines 719-732)

6) The experimental validation of this dataset is limited to showing that CD4 T cells in persistent infection express more EOMES than T cells in acutely infected mice and that they express lower levels of THPOK. However, what is the global alignment between flow cytometry data presented in Figure 1 and the scRNAseq data? Were any of the cluster frequencies predicted by the scRNAseq data validated using a protein panel?

We thank the reviewer for giving us the opportunity to explain in more details the steps we take to validate the robustness of our results. The manuscript described the validation of both the presence of the Thpok^low^ Eomes^high^ population (Figure 2E-F and Figure 3—figure supplement 1D) and the decrease of CCR7-expressing cells in chronic settings (Figure 3—figure supplement 1B). In addition, we confirmed by flow cytometry the presence and proportions of major cell populations described in our study (see validation below). It is however important to note that identifying robust T cell states by flow cytometry is difficult and often fails to discriminate T cell subsets from plastic T cell populations. For example, defining Tcm populations by flow cytometry remains challenging as this cell population can only be distinguished from Tfh based on CCR7 expression. With these limitations in mind, our study was precisely designed to overcome these difficulties and define more robustly T cell states by relying on global transcriptional features instead of expression of a few proteins by flow cytometry, which is often highly variable across laboratories, models and tissues.

To directly address the reviewer’s comment, we included our flow cytometry validation using our flow cytometry datasets based on the expression of known markers on virus-specific T cells (new Figure 1—figure supplement 1E). Overall, this analysis shows a very good correspondence between scRNA-seq and flow cytometry in the presence and distribution of the major clusters across conditions. These results are also consistent with previous studies during acute (Ciucci et al., Immunity 2019, PMID: 30638736, Künzli et al., Science Immunology 2020, PMID: 32144185, Marshall et al., Immunity 2011; PMID: 22018471) and chronic (Snell et al., Nature Immunology 2022, PMID: 34795443, Zander et al., Immunity 2022, PMID: 35216666) LCMV infections.

[Editors' note: further revisions were suggested prior to acceptance, as described below.]

The manuscript has been improved but there is one remaining issue that needs to be addressed, as outlined below:Reviewer #1 (Recommendations for the authors):In general this reviewer is satisfied with the revisions to the manuscript, however, one point needs to be better clarified. When the tumor-specific CD4 T cells from TILs were projected into the reference map 40-50% of them mapped into Th1 effectors. Yet upon further analysis and reclustering, these cells ended up being a completely distinct population of cells from the viral effector Th1. Thus, this reviewer is worried this could lead to misinterpretation and incorrect identification of subsets when using the reference map on other systems with unrepresented subsets not in the reference map. Could the authors comment on/clarify this point? It would be helpful to discuss the additional steps needed to verify that the corresponding states determined from projecting one's data into the reference map have similar gene profiles, and if they do not, how to address and identify these novel populations not represented in the map.

This is an important point. In fact, the analysis of T cells from tumor was included in the revised manuscript to illustrate how to detect novel populations in a query dataset compared to the reference. We have at least four possible strategies, all implemented as functions in the ProjecTILs analysis package, to aid the user in detecting new/unrepresented states. First, on a qualitative level, one can inspect the radar plots (function *plot.states.radar()*) to compare the expression of panels of key genes between query and reference. While for all other well-represented cell types the profiles displayed a good correspondence, Th1_Effector cells in the tumor lacked the expression of key Th1 markers (Ccl5, Ly6c2) (Figure 7C). Second, the user can recalculate the UMAP embeddings of the combined reference and query space (function *recalculate.embeddings()*) to assess whether part of the projected data form a novel, separate cluster (e.g. Figure 7D). Third, per-subtype differential gene expression analysis (function *find.discriminant.genes()*) can reveal which and how many genes are differentially expressed between reference and query in a given subtype. In the case of the TIL data, tumoral T helpers showed increased expression of Th2 markers, guiding the users towards interpreting the nature of the novel cluster (Figure 7E). Finally, to provide a more quantitative measure of deviation from the reference, we implemented an average silhouette coefficient per subtype (function *compute_silhouette()*), which aims at measuring the average distance of query cells from their own assigned cluster/state compared to all other clusters/states of the reference. We report below the silhouette scores, normalized by the silhouettes of the reference, for the T cells from tumor and lymphnode from Figure 7, as well as from the dataset from Khatun et al. *JEM* (2020), which come from the same viral model and should act as a control.

The outlier value in this analysis is the normalized silhouette score of tumor-specific cells classified as Th1 effector, indicating that they do match poorly with the Th1 effectors of the reference. Taken together with the other three diagnostic analyses outlined above and described in the manuscript, these results should warn the user of the presence of a novel state in the query dataset.

To better convey this message in the manuscript, we added the following text to the section “Reference map projection to explore CD4^+^ T cell diversity beyond viral infections”:

“While reference maps aim at being as comprehensive as possible, it is possible that new datasets contain novel states that are not represented in the reference, especially when used in different diseases models. In these cases, the user is encouraged to make use of all the analytic tools we provide with the ProjecTILs package (see Methods) to evaluate the degree of correspondence between reference and query, as we illustrated in the case of tumor-specific T cells (Figure 7C-E). These analyses demonstrate the feasibility of using a reference map to describe cell diversity beyond the states already present in the map, and as a strategy to expand references to incorporate novel, unrepresented cell states.”

And a more detailed description of the available analytical functions in a new section in the Methods:

“Detection of novel/unrepresented states

Several utilities are available in the ProjecTILs package to evaluate query projection accuracy and to detect the presence of novel cell states. First, on a qualitative level, one can apply the *plot.states.radar()* function to compare the expression of panels of key genes between query and reference. Second, the user can recalculate the UMAP embeddings of the combined reference and query space (function *recalculate.embeddings()*) to assess whether part of the projected data form a new, separate cluster (e.g. Figure 7D). Third, per-subtype differential expression analysis (function *find.discriminant.genes()*) can reveal which and how many genes are differentially expressed between reference and query in a given subtype (e.g. Figure 7E). Fourth, the *compute_silhouette()* function calculates an average silhouette coefficient per subtype, which aims at measuring the average distance of query cells from their own assigned cluster compared to all other clusters of the reference. A case study highlighting the application of these metrics can be found at: https://carmonalab.github.io/ProjecTILs_CaseStudies/novelstate.html”